# Supervised Community Detection with Line Graph Neural Networks

**Zhengdao Chen**
Courant Institute of Mathematical Sciences
New York University, New York, NY

**Lisha Li**
Amplify Partners
San Francisco, CA

**Joan Bruna**
Courant Institute of Mathematical Sciences
New York University, New York, NY

## Abstract

Community detection in graphs can be solved via spectral methods or posterior inference under certain probabilistic graphical models. Focusing on random graph families such as the stochastic block model, recent research has unified both approaches and identified both statistical and computational detection thresholds in terms of the signal-to-noise ratio. By recasting community detection as a node-wise classification problem on graphs, we can also study it from a learning perspective. We present a novel family of Graph Neural Networks (GNNs) for solving community detection problems in a supervised learning setting. We show that, in a data-driven manner and without access to the underlying generative models, they can match or even surpass the performance of the belief propagation algorithm on binary and multiclass stochastic block models, which is believed to reach the computational threshold in these cases. In particular, we propose to augment GNNs with the non-backtracking operator defined on the line graph of edge adjacencies. The GNNs are achieved good performance on real-world datasets. In addition, we perform the first analysis of the optimization landscape of using (linear) GNNs to solve community detection problems, demonstrating that under certain simplifications and assumptions, the loss value at any local minimum is close to the loss value at the global minimum/minima.

## 1 Introduction

Graph inference problems encompass a large class of tasks and domains, from posterior inference in probabilistic graphical models to community detection and ranking in generic networks, image segmentation or compressed sensing on non-Euclidean domains. They are motivated both by practical applications, such as in the case of PageRank, and also by fundamental questions on the algorithmic hardness of solving such tasks.

From a data-driven perspective, these problems can be formulated in unsupervised, semi-supervised or supervised learning settings. In the supervised case, one assumes a dataset of graphs with labels on their nodes, edges or the entire graphs, and attempts to perform node-wise, edge-wise and graph-wise classification by optimizing a loss over a certain parametric class, e.g. neural networks. Graph Neural Networks (GNNs) are natural extensions of Convolutional Neural Networks to graph-structured data, and have emerged as a powerful class of algorithms to perform complex graph inference leveraging labeled data (Gori et al., 2005; Bronstein et al., 2017) (and references therein). In essence, these neural networks learn cascaded linear combinations of intrinsic graph operators interleaved with node-wise (or edge-wise) activation functions. Since they utilize intrinsic graph operators, they can be applied to varying input graphs, and they offer the same parameter sharing advantages as their CNN counterparts.

In this work, we focus on community detection problems, a wide class of node classification tasks that attempt to discover a clustered, segmented structure within a graph. The algorithmic approaches to this problem include a rich class of spectral methods, which take advantage of the spectrum of

certain operators defined on the graph, as well as approximate message-passing methods such as belief propagation (BP), which performs approximate posterior inference under predefined graphical models (Decelle et al., 2011). Focusing on the supervised setting, we study the ability of GNNs to approximate, generalize or even improve upon these class of algorithms. Our motivation is two-fold. On the one hand, this problem exhibits algorithmic hardness on some settings, opening up the possibility to discover more efficient algorithms than the current ones. On the other hand, many practical scenarios fall beyond pre-specified probabilistic models, requiring data-driven solutions.

We propose modifications to the GNN architecture, which allow it to exploit edge adjacency information, by incorporating the non-backtracking operator of the graph. This operator is defined over the edges of the graph and allows a directed flow of information even when the original graph is undirected. It was introduced to community detection problems by Krzakala et al. (2013), who propose a spectral method based on the non-backtracking operator. We refer to the resulting GNN model as a *Line Graph Neural Network (LGNN)*. Focusing on important random graph families exhibiting community structure, such as the stochastic block model (SBM) and the geometric block model (GBM), we demonstrate improvements in the performance by our GNN and LGNN models compared to other methods, including BP, even in regimes within the so-called computational-to-statistical gap. A perhaps surprising aspect is that these gains can be obtained even with *linear* LGNNs, which become parametric versions of power iteration algorithms.

We want to mention that besides community detection tasks, GNN and LGNN can be applied to other node-wise classification problems too. The reason we are focusing on community detection problems is that this is a relatively well-studied setup, for which different algorithms have been proposed and where computational and statistical thresholds have been studied in several scenarios. Moreover, synthetic datasets can be easily generated for community detection tasks. Therefore, we think it is a nice setup for comparing different algorithms, besides its practical values.

The good performances of GNN and LGNN motivate our second main contribution: the analysis of the optimization landscape of simplified and linear GNN models when trained with planted solutions of a given graph distribution. Under reparametrization, we provide an upper bound on the *energy gap* controlling the energy difference between local and global minima (or minimum). With some assumptions on the spectral concentration of certain random matrices, this energy gap will shrink as the size of the input graphs increases, which would mean that the optimization landscape is benign on large enough graphs.

**Summary of Main Contributions:**

- We propose an extension of GNNs that operate on the line graph using the non-backtracking operator, which yields improvements on hard community detection regimes.

- We show that on the stochastic block model we reach detection thresholds in a purely data-driven fashion, in the sense that our results improve upon belief propagation in hard SBM detection regimes, as well as in the geometric block model.

- We perform the first analysis of the learning landscape of GNN models, showing that under certain simplifications and assumptions, they exhibit a form of "energy gap", where local mimima are confined in low-energy configurations.

- We show that our model can perform well on community detection problems with real-world datasets.

## 2 PROBLEM SETUP

We are interested in a specific class of node-classification tasks in which given an input graph $G = (V, E)$, a labeling $y : V \rightarrow \{1, \ldots, C\}$ that encodes a partition of $V$ into $C$ communities is to be predicted at each node. We assume that a training set $\{(G_t, y_t)\}_{t \leq T}$ is given, with which we train a model that predicts $\hat{y} = \Phi(G, \theta)$ by minimizing

$$L(\theta) = \frac{1}{T} \sum_{t \leq T} \ell(\Phi(G_t, \theta), y_t)$$

Since $y$ encodes a partition of $C$ groups, the specific label of each node is only important up to a global permutation of $\{1, \ldots, C\}$. Section 4.3 describes how to construct loss functions $\ell$ with such a property. A permutation of the observed nodes translates into the same permutation applied to the labels, which justifies models $\Phi$ that are equivariant to permutations. Also, we are interested in inferring properties of community detection algorithms that do not depend on the specific size of the graphs[1]. We therefore require that the model $\Phi$ accepts graphs of variable size for the same set of parameters, similar to sequential RNN or spatial CNN models.

## 3 RELATED WORK

GNN was first proposed in Gori et al. (2005); Scarselli et al. (2009). Bruna et al. (2013) generalize convolutional neural networks on general undirected graphs by using the graph Laplacian's eigenbasis. This was the first time the Laplacian operator was used in a neural network architecture to perform classification on graph inputs. Defferrard et al. (2016) consider a symmetric Laplacian generator to define a multiscale GNN architecture, demonstrated on classification tasks. Similarly, Kipf & Welling (2016) use a similar generator as effective embedding mechanisms for graph signals and applies it to semi-supervised tasks. This is the closest application of GNNs to our current contribution. However, we highlight that semi-supervised learning requires bootstrapping the estimation with a subset of labeled nodes, and is mainly interested in generalization within a single, fixed graph. In comparison, our setup considers community detection across a distribution of input graphs and assumes no initial labeling on the graphs in the test dataset except for the adjacency information.

There have been several extensions of GNNs by modifying their non-linear activation functions, parameter sharing strategies, and choice of graph operators (Li et al., 2015; Sukhbaatar et al., 2016; Duvenaud et al., 2015; Niepert et al., 2016). In particular, Gilmer et al. (2017) interpret the GNN architecture as learning an approximate message-passing algorithm, which extends the learning of hidden representations to graph edges in addition to graph nodes. Recently, Velickovic et al. (2017) relate adjacency learning with attention mechanisms, and Vaswani et al. (2017) propose a similar architecture in the context of machine translation. Another recent and related piece of work is by Kondor et al. (2018), who propose a generalization of GNN that captures high-order node interactions through covariant tensor algebra. Our approach to extend the expressive power of GNN using the line graph may be seen as an alternative to capture such high-order interactions.

Our energy landscape analysis is related to the recent paper by Shamir (2018), which establishes an energy bound on the local minima arising in the optimization of ResNets. In our case, we exploit the properties of the community detection problem to produce an energy bound that depends on the concentration of certain random matrices, which one may hope for as the size of the input graphs increases. Finally, Zhang (2016)'s work on data regularization for clustering and rank estimation is also motivated by the success of using Bethe-Hessian-like perturbations to improve spectral methods on sparse networks. It finds good perturbations via matrix perturbations and also has successes on the stochastic block model. Yang & Leskovec (2012a) curate benchmark datasets for community detection and quantify the quality of these datasets, while Yang & Leskovec (2012b) develop new algorithms for community detection by fitting to the networks the Affliation Graph Model (AGM), a generative model for graphs with overlapping communities.

## 4 LINE GRAPH NEURAL NETWORKS

This section introduces our GNN architectures that include the power graph adjacency (Section 4.1) and its extension to line graphs using the non-backtracking operator (Section 4.2), as well as the design of losses invariant to global label permutations (Section 4.3).

### 4.1 GRAPH NEURAL NETWORKS USING A FAMILY OF GRAPH OPERATORS

The Graph Neural Network (GNN), introduced in Scarselli et al. (2009) and later simplified in Li et al. (2015); Duvenaud et al. (2015); Sukhbaatar et al. (2016), is a flexible neural network architecture based on local operators on a graph $G = (V, E)$. Given a state vector $x \in \mathbb{R}^{|V| \times b}$ on the vertices of

---

[1]In this work, however, we assume that $C$ is fixed.

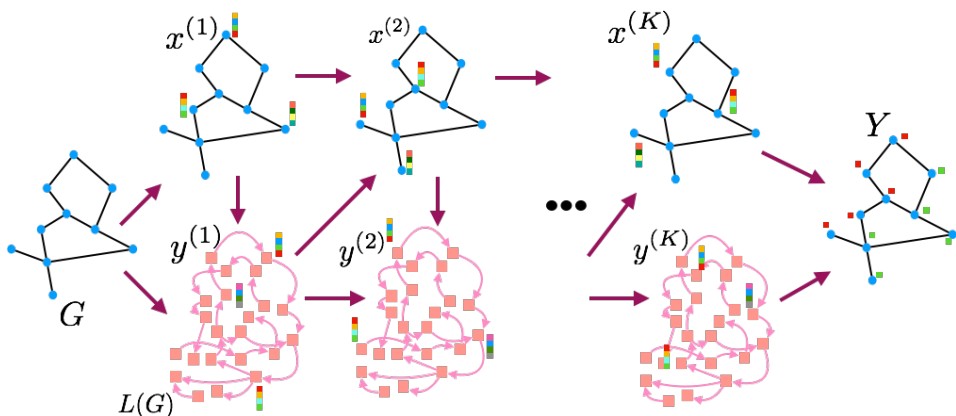

Figure 1. Overview of the architecture of LGNN (Section 4.2). Given a graph $G$, we construct its line graph $L(G)$ with the non-backtracking operator (Figure 2). In every layer, the states of all nodes in $G$ and $L(G)$ are updated according to (2). The final states of nodes in G are used to predict node-wise labels, and the trainining is performed end-to-end using standard backpropagation with a label permutation invariant loss (Section 4.3).

$G$, we consider intrinsic linear operators of the graph that act locally on $x$, which can be represented as $|V|$-by-$|V|$ matrices. For example, the *adjacency matrix* $A$ is defined entry-wise by $A_{i_1 i_2} = 1$ if $(i_1, i_2) \in E$ and $A_{i_1 i_2} = 0$ if $(i_1, i_2) \notin E$, for every pair $(i_1, i_2) \in V \times V$. The *degree matrix* $D$ is then defined as $\text{diag}(A\mathbb{1})$, i.e., $D$ is a diagonal matrix with $D_{ii}$ being the number of edges that the $i$th node has. We can also define *power graph adjacency matrices* as $A_{(j)} = \min(1, A^{2^j})$, which encodes $2^j$-hop neighborhoods into a binary graph. Finally, there is also the *identity matrix*, $I$. Given such a family of operators for each graph, $\mathcal{F}_A^J = \{I, D, A, A_{(2)}, ..., A_{(J)}\}$, we define a GNN layer that maps $x^{(k)} \in \mathbb{R}^{|V| \times b_k}$ to $x^{(k+1)} \in \mathbb{R}^{|V| \times b_{k+1}}$ as follows. First, we compute

$$z^{(k+1)} = \rho \left[ \sum_{O_i \in \mathcal{F}_A^J} O_i x^{(k)} \theta_i \right], \overline{z}^{(k+1)} = \sum_{O_i \in \mathcal{F}_A^J} O_i x^{(k)} \theta_i \tag{1}$$

where $\theta_j \in \mathbb{R}^{b_k \times \frac{b_{k+1}}{2}}$ are trainable parameters and $\rho(\cdot)$ is a point-wise nonlinear activation function, chosen in this work to be the ReLU function, i.e. $\rho(z) = \max(0, z)$ for $z \in \mathbb{R}$. Then we define $x^{(k+1)} = [z^{(k+1)}, \overline{z}^{(k+1)}] \in \mathbb{R}^{|V| \times b_{k+1}}$ as the concatenation of $z^{(k+1)}$ and $\overline{z}^{(k+1)}$. The layer thus includes linear "residual connections" (He et al., 2016) via $\overline{z}^{(k)}$, both to ease with the optimization when using large number of layers and to increase the expressivity of the model by enabling it to perform power iterations. Since the spectral radius of the learned linear operators in (1) can grow as the optimization progresses, the cascade of GNN layers can become unstable to training. In order to mitigate this effect, we consider spatial batch normalization (Ioffe & Szegedy, 2015) at each layer.[2] In our experiments, the initial states are set to be the degrees of the nodes, i.e., $x^{(0)} = \deg(A) := A\mathbb{1}$. In addition, the model $\Phi(G, x^{(0)}) = x^{(K)}$ satisfies the permutation equivariance property required for community detection: Given a permutation $\pi$ among the nodes in the graph, $\Phi(G_\pi, \Pi x^{(0)}) = \Pi \Phi(G, x^{(0)})$, where $\Pi$ is the permutation matrix associated with $\pi$.

**Analogy between GNN and power iterations** In our setup, spatial batch normalization not only prevents gradient blowup, but also performs the orthogonalisation relative to the constant vector, which reinforces the analogy with the spectral methods for community detection, some background of which is described in B.1. In essence, in certain regimes, the eigenvector of $A$ corresponding to its second largest eigenvalue and the eigenvector of the *Laplacian matrix*, $L = D - A$, corresponding to its second smallest eigenvalue (i.e. the Fiedler vector), are both correlated with the community structure of the graph. Thus, spectral methods for community detection performs power iterations on

---

[2]The term "spatial batch normalization" may be slightly misleading, since each batch only contain one graph in our experiments. However, we used spatial batch normalization in our implementation for convenience, as it performs orthogonalization and normalization, as further explained in the next paragraph.

these matrices to obtain the eigenvectors of interest and predicts the community structure based on them. For example, to extract the Fiedler vector of a matrix $M$, whose eigenvector corresponding to the smallest eigenvalue is known to be $v$, one performing power iterations on $\tilde{M} = \|M\|I - M$ by $y^{(n+1)} = \tilde{M}x^{(n)}$, $x^{(n+1)} = \frac{y^{(n+1)} - v^T vy^{(n+1)}}{\|y^{(n+1)} - v^T vy^{(n+1)}\|}$. If $v$ is a constant vector, which is the case for $L$, then the normalization above is precisely performed within the spatial batch normalization step.

By incorporating a family of operators into the neural network framework, the GNN can not only approximate but also go beyond power iterations. As explained in Section B.1, the Krylov subspace generated by the graph Laplacian (Defferrard et al., 2016) is not sufficient to operate well in the sparse regime, as opposed to the space generated by $\{I, D, A\}$. The expressive power of each layer is further increased by adding multiscale versions of $A$, although this benefit comes at the cost of computational efficiency, especially in the sparse regime. The network depth is chosen to be of the order of the graph diameter, so that all nodes obtain information from the entire graph. In sparse graphs with small diameter, this architecture offers excellent scalability and computational complexity. Indeed, in many social networks diameters are constant (due to hubs), or $\log(|V|)$, as in the stochastic block model in the constant average degree regime (Riordan & Wormald, 2010). This results in a model with computational complexity on the order of $|V|\log(|V|)$, making it amenable to large-scale graphs.

## 4.2 LGNN: GNN ON LINE GRAPHS WITH THE NON-BACKTRACKING OPERATOR

For graphs with few cycles, posterior inference can be remarkably approximated by loopy belief propagation (Yedidia et al., 2003). As described in Section B.2, the message-passing rules are defined over the edge adjacency graph (see equation 57). Although its second-order approximation around the critical point can be efficiently approximated with a power method over the original graph, a data-driven version of BP requires accounting for the non-backtracking structure of the message-passing. In this section we describe an upgraded GNN model that exploits the non-backtracking structure.

The line graph $L(G) = (V_L, E_L)$ is a graph representing the edge adjacency structure of $G$. If $G = (V, E)$ is an undirected graph, then the vertices $V_L$ of $L(G)$ are the ordered edges in $E$, i.e., $V_L = \{(i \to j) : (i, j) \in E\}$, and so $|V_L| = 2|E|$. The non-backtracking operator on the line graph is represented by a matrix $B \in \mathbb{R}^{2|E| \times 2|E|}$ defined as

$$B_{(i \to j),(i' \to j')} = \begin{cases} 1 & \text{if } j = i' \text{ and } j' \neq i, \\ 0 & \text{otherwise.} \end{cases}$$

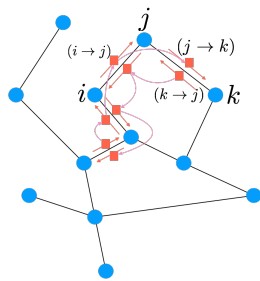

Figure 2. Construction of the line graph $L(G)$ using the non-backtracking Operator. The nodes of $L(G)$ correspond to oriented edges of $G$.

This operator enables the directed propagation of information through on the line graph and was first proposed in the context of community detection on sparse graphs in Krzakala et al. (2013). The message-passing rules of BP can be expressed as a diffusion in the line graph $L(G)$ using this non-backtracking operator, with specific choices of activation function that turn product of beliefs into sums.

A natural extension of the GNN architecture presented in Section 4.1 is thus to consider a second GNN defined on $L(G)$, where $B$ and $D_B = \text{diag}(B\mathbb{1})$ play the role of the adjacency and the degree matrices, respectively. This effectively defines *edge features* that are updated according to the edge adjacency of $G$. Edge and node features communicate at each layer using the edge indicator matrices $P_m, P_d \in \{0,1\}^{|V| \times 2|E|}$, defined as $P_{m\,i,(i \to j)} = 1$, $P_{d\,j,(i \to j)} = 1$, $P_{d\,i,(i \to j)} = 1$, $P_{d\,j,(i \to j)} = -1$ and $0$ otherwise. With the skip linear connections defined similarly, the resulting model becomes

$$z^{(k+1)} = \rho \left[ \sum_{O_i \in \mathcal{F}_A} O_i x^{(k)} \theta_i + \sum_{O'_j \in \mathcal{F}_{AB}} O'_j y^{(k)} \theta'_i \right]$$

$$w^{(k+1)} = \rho \left[ \sum_{O''_l \in \mathcal{F}_B} O''_l y^{(k)} \theta''_i + \sum_{O'_j \in \mathcal{F}_{AB}} (O'_j)^T x^{(k+1)} \theta'''_j \right]$$

(2)

where $\mathcal{F}_A = \{I, D, A, A_{(2)}, \ldots, A_{(J)}\}$, $\mathcal{F}_B = \{I_B, D_B, B, B_{(2)}, \ldots, B_{(J)}\}$, $\mathcal{F}_{AB} = \{P_m, P_d\}$, and the trainable parameters are $\theta_i, \theta_i', \theta_i'' \in \mathbb{R}^{b_k \times b_{k+1}}$ and $\theta_i''' \in \mathbb{R}^{b_{k+1} \times b_{k+1}}$. We call such a model a *Line Graph Neural Network* (LGNN).

In our experiments, we set $x^{(0)} = \deg(A)$ and $y^{(0)} = \deg(B)$. For graph families with constant average degree $\overline{d}$ (as $|V|$ grows), the line graph has size $2|E| \sim O(\overline{d}|V|)$, and is therefore feasible from the computational point of view. Furthermore, the construction of line graphs can be iterated to generate $L(L(G))$, $L(L(L(G)))$, etc. to yield a *graph hierarchy*, which could capture high-order interactions among nodes of $G$. Such an hierarchical construction is related to other recent efforts to generalize GNNs (Kondor et al., 2018).

**Relationship between LGNN and edge feature learning approaches**   Several authors have proposed combining node and edge feature learning. Battaglia et al. (2016) introduce edge features over directed and typed graphs, but does not discuss the undirected case. Kearnes et al. (2016); Gilmer et al. (2017) learn edge features on undirected graphs using $f_e = g(x(i), x(j))$ for an edge $e = (i, j)$, where $g$ is commutative on its arguments. Finally, Velickovic et al. (2017) learns directed edge features on undirected graphs using stochastic matrices as adjacencies (which are either row or column-normalized). However, we are not aware of works that consider the edge adjacency structure provided by the non-backtracking matrix on the line graph. With non-backtracking matrix, our LGNN can be interpreted as learning *directed* edge features from an *undirected* graph. Indeed, if each node $i$ contains two distinct sets of features $x_s(i)$ and $x_r(i)$, the non-backtracking operator constructs edge features from node features while preserving orientation: For an edge $e = (i, j)$, our model is equivalent to constructing oriented edge features $f_{i \to j} = g(x_s(i), x_r(j))$ and $f_{j \to i} = g(x_r(i), x_s(j))$ (where $g$ is trainable and not necessarily commutative on its arguments) that are subsequently propagated through the graph. Constructing such local oriented structure is shown to be important for improving performance in the next section.

For comparison, we also define a *linear LGNN (LGNN-L)* as the the LGNN that drops the nonlinear activation functions $\rho$ in (2), and a *symmetric LGNN (LGNN-S)* as the LGNN whose line graph is defined on the *undirected* edges of the original graph: In LGNN-S, two edges of $G$ are connected in the line graph if and only if they share one common node; also, $\mathcal{F} = \{P\}$, with $P \in \mathbb{R}^{|V| \times |E|}$ defined as $P_{i,(j \to k)} = 1$ if $i = j$ or $k$ and $0$ otherwise.

## 4.3   A Loss Function Invariant Uner Label permutation

Let $\mathcal{C} = \{1, \ldots, C\}$ denote the set of all community labels, and consider first the case where communities do not overlap. By applying the softmax function at the end, we interpret the $c$th dimension of the output of the models at node $i$ as the conditional probability that the node belongs to community $c$, $o_{i,c} = p(y_i = c \,|\, \theta, G)$. Let $G = (V, E)$ be the input graph and let $y_i$ be the ground truth community label of node $i$. Since the community structure is defined up to global permutations of the labels, we can define a loss function with respect to a given graph instance as

$$\ell(\theta) = \inf_{\pi \in S_\mathcal{C}} -\sum_{i \in V} \log o_{i, \pi(y_i)} \,, \tag{3}$$

where $S_\mathcal{C}$ denotes the permutation group of $C$ elements. This is essentially taking the the cross entropy loss minimized over all possible permutations of $\mathcal{C}$. In our experiments, we considered examples with small numbers of communities such as 2 and 5. In general scenarios where $C$ is much larger, the evaluation of the loss function (3) can be impractical due to the minimization over $S_\mathcal{C}$. A possible solution is to randomly partition $C$ labels into $\tilde{C}$ groups, and then to marginalize the model outputs $o_{i,c}$, $c \leq C$ into $\bar{o}_{i,\bar{c}} = \sum_{c \in \bar{c}} o_{i,c}$, $\bar{c} \in \tilde{C}$, and and finally use $\ell(\theta) = \inf_{\pi \in S_{\tilde{C}}} -\sum_{i \in V} \log \bar{o}_{i, \pi(\bar{y}_i)}$ as an approximate loss value, which only involves a permutation group of size $(\tilde{C}!)$.

Finally, if communities may overlap, we can enlarge $\mathcal{C}$ to include subsets of communities and define the permutation group accordingly. For example, if there are two overlapping communities, we let $\mathcal{C} = \{\{1\}, \{2\}, \{1, 2\}\}$, and only allow the permutation between 1 and 2 when computing the loss function as well as the overlap to be introduced in Section 6.

## 5 ENERGY LANDSCAPE OF LINEAR GNN OPTIMIZATION

As described in the numerical experiments, we found that the GNN models without nonlinear activations already provide substantial gains relative to baseline (non-trainable) algorithms. This section studies the optimization landscape of linear GNNs. Despite defining a non-convex objective, we prove that the landscape is "benign" under certain further simplifications, in the sense that the local minima are confined in sublevel sets of low energy.

For simplicity, we consider only the binary $c = 2$ case where we replace the node-wise binary cross-entropy loss by the squared cosine distance[3], assume a single feature map ($b_k = 1$ for all $k$), and focus on the GNN described in Section 4.1 (although our analysis carries equally to describe the line graph version; see remarks below). We also make the simplifying assumption to replace the layer-wise spatial batch normalization by a simpler projection onto the unit $\ell_2$ ball (thus we do not remove the mean). Without loss of generality, assume that the input graph $G$ has size $n$, and denote by $\mathcal{F} = \{A_1, \ldots, A_Q\}$ the family of graph operators appearing in (1). Each layer thus applies an arbitrary polynomial $\sum_{q=1}^{Q} \theta_q^{(k)} A_q$ to the incoming node feature vector $x^{(k)}$. Given an input node vector $w \in \mathbb{R}^n$, the network output can thus be written as

$$\hat{Y} = \frac{e}{\|e\|} \, , \text{ with } e = \left( \prod_{k=1}^{K} \sum_{q \leq Q} \theta_q^{(k)} A_q \right) w \, . \tag{4}$$

We highlight that this linear GNN setup is fundamentally different from the linear fully-connected neural networks (that is, neural networks with linear activation function), whose landscape has been analyzed in Kawaguchi (2016). First, the output of the GNN is on the unit sphere, which has a different geometry. Next, since the operators in $\mathcal{F}$ depend on the input graph, they introduce fluctuations in the landscape. In general, the operators in $\mathcal{F}$ are not commutative, but by considering the generalized Krylov subspace generated by powers of $\mathcal{F}$, $\mathcal{F}^K = \{O_1 = A_1^K, O_2 = A_1 A_2^{K-1}, O_3 = A_1 A_2 A_1^{K-2}, \ldots O_{Q^K} = A_Q^K\}$, one can reparametrize (4) as $e = \sum_{j=1}^{Q^K} \beta_j O_j w$ with $\beta \in \mathbb{R}^M$, with $M = Q^K$. Given the target $y \in \mathbb{R}^n$, the loss incurred by each pair $(G, y)$ becomes $1 - \frac{|\langle e, y \rangle|^2}{\|e\|^2}$, and therefore the population loss, when expressed in terms of $\beta$, equals

$$L_n(\beta) = 1 - \mathbb{E}_{X_n, Y_n} \frac{\beta^\top Y_n \beta}{\beta^\top X_n \beta} \, , \text{ with} \tag{5}$$

$$Y_n = z_n z_n^\top \in \mathbb{R}^{M \times M} \, , \, (z_n)_j = \langle O_j w, y \rangle \text{ and } X_n = U_n U_n^\top \in \mathbb{R}^{M \times M} \, , U_n = \begin{bmatrix} (O_1 w)^\top \\ \cdots \\ (O_M w)^\top \end{bmatrix} \, .$$

The landscape is thus specified by a pair of random matrices $Y_n, X_n \in \mathbb{R}^{M \times M}$.

Assuming that $\mathbb{E}X_n \succ 0$, we write the Cholesky decomposition of $\mathbb{E}X_n$ as $\mathbb{E}X_n = R_n R_n^T$, and define $A_n = R_n^{-1} Y_n (R_n^{-1})^T$, $\bar{A}_n = \mathbb{E}A_n = R_n^{-1} \mathbb{E}Y_n (R_n^{-1})^T$, $B_n = R_n^{-1} X_n (R_n^{-1})^T$, and $\Delta B_n = B_n - I_n$. Given a symmetric matrix $K \in \mathbb{R}^{M \times M}$, we let $\lambda_1(K), \lambda_2(K), ..., \lambda_M(K)$ denote the eigenvalues of $K$ in nondecreasing order. Then, the following theorem establishes that under appropriate assumptions, the concentration of relevant random matrices around their mean controls the energy gaps between local and global minima of $L$.

**Theorem 5.1.** *For a given $n$, let $\eta_n = (\lambda_1(\bar{A}_n) - \lambda_2(\bar{A}_n))^{-1}$, $\mu_n = \mathbb{E}[|\lambda_1(A_n)|^6]$, $\nu_n = \mathbb{E}[|\lambda_1(B_n)|^{-6}]$, $\delta_n = \mathbb{E}[\|\Delta B_n\|^6]$, and assume that all four quantities are finite. Then if $\beta_l \in \mathbb{S}^{M-1}$ is a local minimum of $L_n$, and $\beta_g \in \mathbb{S}^{M-1}$ is a global minimum of $L_n$, we have $L_n(\beta_l) \leq (1 + \epsilon_{\eta_n, \mu_n, \nu_n, \delta_n}) \cdot L_n(\beta_g)$, where $\epsilon_{\eta_n, \mu_n, \nu_n, \delta_n} = O(\delta_n)$ for given $\eta_n, \mu_n, \nu_n$ as $\delta_n \to 0$ and its formula is given in the appendix.*

**Corollary 5.2.** *If $(\eta_n)_{n \in \mathbb{N}^*}$, $(\mu_n)_{n \in \mathbb{N}^*}$, $(\nu_n)_{n \in \mathbb{N}^*}$ are all bounded sequences, and $\lim_{n \to \infty} \delta_n = 0$, then $\forall \epsilon > 0$, $\exists n_\epsilon$ such that $\forall n > n_\epsilon$, $|L_n(\beta_l) - L_n(\beta_g)| \leq \epsilon \cdot L_n(\beta_g)$.*

The main strategy of the proof is to consider the actual loss function $L_n$ as a perturbation of $\tilde{L}_n(\beta) = 1 - \mathbb{E}_{X_n, Y_n} \frac{\beta^T Y_n \beta}{\beta^T \mathbb{E}X_n \beta} = 1 - \frac{\beta^T \mathbb{E}Y_n \beta}{\beta^T \mathbb{E}X_n \beta}$, which has a landscape that is easier to analyze and

---

[3]to account for the invariance up to global flip of label

does not have poor local minima, since it is equivalent to a quadratic form defined over the sphere $\mathbb{S}^{M-1}$. Applying this theorem requires estimating spectral fluctuations of the pair $X_n, Y_n$, which in turn involve the spectrum of the $C^*$ algebras generated by the non-commutative family $\mathcal{F}$. For example, for stochastic block models, it is an open problem how the bound behaves as a function of the parameters $p$ and $q$. Another interesting question is to understand how the asymptotics of our landscape analysis relate to the hardness of estimation as a function of the signal-to-noise ratio. Finally, another open question is to what extent our result could be extended to the non-linear residual GNN case, perhaps leveraging ideas from Shamir (2018).

## 6    EXPERIMENTS

We present experiments on community detection in synthetic datasets (Sections 6.1, 6.2 and Appendix C.1) as well as real-world datasets (Section 6.3). In the synthetic experiments, the performance is measured by the overlap between predicted ($\hat{y}$) and true labels ($y$), which quantifies how much better than random guessing a predicted labeling is, given by $\left( \frac{1}{n} \sum_u \delta_{y(u),\hat{y}(u)} - \frac{1}{C} \right) / (1 - \frac{1}{C})$, where $\delta$ is the Kronecker delta function, and this quantity is maximized over global permutations within a graph of the set of labels. In the real-world datasets, as the communies are overlapping and not balanced, the prediction accuracy is measured by $\frac{1}{n} \sum_u \delta_{y(u),\hat{y}(u)}$, and the set of permutations to be maximized over is described in Section 4.3. We used Adamax (Kingma & Ba, 2014) with learning rate 0.004 across all experiments. All the neural network models have 30 layers and 8 features in the middle layers (i.e., $b_k = 8$) for experiments in Sections 6.1 and 6.2, and 20 layers and 6 features for Section 6.3. GNNs and LGNNs have $J = 2$ across the experiments except the ablation experiments in Section C.3. [4]

### 6.1    BINARY STOCHASTIC BLOCK MODEL

The stochastic block model is a random graph model with planted community structure. In its simplest form, the graph consists of $|V| = n$ nodes, which are partitioned into $C$ communities, that is, each node is assigned a label $y \in \{1, ..., C\}$. An edge connecting any two vertices $u, v$ is drawn independently at random with probability $p$ if $y(v) = y(u)$, and with probability $q$ otherwise. In the binary case (i.e. $C = 2$), the sparse regime, where $p, q \simeq 1/n$, is well understood and provides an initial platform to compare the GNN and LGNN with provably optimal recovery algorithms (Appendix B). We consider two learning scenarios. In the first scenario, we choose different pairs of $p$ and $q$, and train the models for each pair separately. In particular, for each pair of $(p_i, q_i)$, we sample 6000 graphs under $G \sim SBM(n = 1000, p_i, q_i, C = 2)$ and then train the models for each $i$. In the second scenario, reported in Appendix C.2, we train a single set of parameters $\theta$ from a set of 6000 graphs sampled from a mixture of SBM with different pairs of $(p_i, q_i)$, and average degree. Importantly, his setup shows that our models are not simply approximating known algorithms such as BP for particular SBM parameters, since the parameters vary in this dataset.

For the first scenario, we chose five different pairs of $(p_i, q_i)$ while fixing $p_i + q_i$, thereby corresponding to different signal-to-noise ratios (SNRs). Figure 3 reports the performance of our models on the binary SBM model for the different SNRs, compared with baseline methods including BP, spectral methods using the normalized Laplacian and the Bethe Hessian as well as Graph Attention Networks (GAT)[5] from Velickovic et al. (2017). We observe that both GNN and LGNN reach the performance of BP. In addition, even the linear LGNN achieves a performance that is quite close to that of BP, in accordance to the spectral approximations of BP given by the Bethe Hessian (see supplementary), and significantly outperforms performing 30 power iterations on the Bethe Hessian or the normalized Laplacian, as was done in the spectral methods. We also notice that our models outperform GAT in this task. We ran experiments in the dissociative case ($q > p$), as well as with $C = 3$ communities and obtained similar results, which are not reported here.

---

[4]Code is available at `https://github.com/zhengdao-chen/GNN4CD`

[5]Our implementation of GAT is based on `https://github.com/Diego999/pyGAT`. We modified the code so that the number of layers in the network is flexible, and also added spatial batch normalization at the end of each layer, similar to in our GNN and LGNN, as our experiments showed that including spatial batch normalization improved the performance. The results in sections 6.1 and 6.2 are from a trained GAT with 30 layers and 8 features.

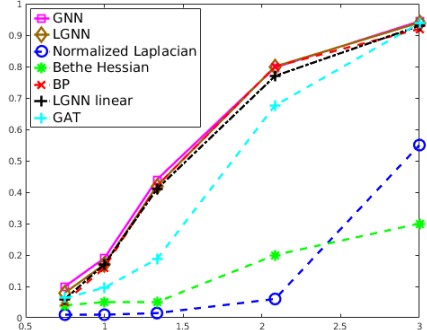

Figure 3. Binary associative SBM detection ($C = 2$, p > q). X-axis corresponds to SNR, and Y-axis to overlap between the prediction and the ground truth.

|  | GNN | LGNN | LGNN-L | LGNN-S | GAT | BP |
|---|---|---|---|---|---|---|
| Avg. | 0.18 | 0.21 | 0.18 | 0.18 | 0.16 | 0.14 |
| Std. Dev. | 0.04 | 0.05 | 0.04 | 0.04 | 0.04 | 0.02 |

Table 1: Performance of different models on 5-community dissociative SBM graphs with $n = 400$, $C = 5$, $p = 0$, $q = 18/n$, corresponding to an average degree of 14.5. The first row gives the average overlap across test graphs, and the second row gives the graph-wise standard deviation of the overlap.

## 6.2 COMPUTATIONAL-TO-STATISTICAL THRESHOLDS IN THE SBM

In SBM with fewer than 4 communities, it is known that BP provably reaches the information-theoretic threshold (Abbe, 2017; Massoulié, 2014; Coja-Oghlan et al., 2016). The situation is different for $k > 4$, where it is conjectured that a gap emerges between the theoretical performance of MLE estimators and the performance of any polynomial-time estimation procedure (Decelle et al., 2011). In this context, one can use the GNN models to search the space of the generalizations of BP, and attempt to improve upon the detection performance of BP for scenarios where the SNR falls within the computational-to-statistical gap. Table 1 presents results for the 5-community dissociative SBM, with $n = 400$, $p = 0$ and $q = 18/n$. The SNR in this setup is above the information-theoretic threshold but below the asymptotic threshold above which BP is able to detect (Decelle et al., 2011). Note that since $p = 0$, this also amounts to a graph coloring problem.

We see that the GNN and LGNN models outperform BP in this experiment, indeed opening up the possibility to reduce the computation-information gap. That said, our model may taking advantage of finite-size effects, which will vanish as $n \to \infty$. The asymptotic study of these gains is left for future work. In terms of average test accuracy, LGNN has the best performance. In particular, it outperforms the symmetric version of LGNN, emphasizing the importance of the non-backtracking matrix used in LGNN. Although equipped with the attention mechanism, GAT does not explicitly incorporate in itself the degree matrix, the power graph adjacency matrices or the line graph structure, and has inferior performance compared with the GNN and LGNN models. Further ablation studies on GNN and LGNN are described in Section C.3.

## 6.3 REAL DATASETS FROM SNAP

We now compare the models on the SNAP datasets, whose domains range from social networks to hierarchical co-purchasing networks. We obtain the training set as follows. For each SNAP dataset, we start by focusing only on the 5000 top quality communities provided by the dataset. We then identify edges $(i, j)$ that cross at least two different communities. For each of such edges, we consider pairs of communities $C_1, C_2$ such that $i \notin C_2$ and $j \notin C_1$, $i \in C_1$, $j \in C_2$, and extract the subset of nodes determined by $C_1 \cup C_2$ together with the edges among them. The resulting graph is connected since each community is connected. Finally, we divide the dataset into training and testing sets by enforcing that no community belongs to both the training and the testing set. In our experiment, due to computational limitations, we restrict our attention to the three smallest datasets in the SNAP

collection (Youtube, DBLP and Amazon), and we restrict the largest community size to 200 nodes, which is a conservative bound.

We compare the performance of GNN and LGNN models with GAT as well as the Community-Affiliation Graph Model (AGM), which is a generative model proposed in Yang & Leskovec (2012b) that captures the overlapping structure of real-world networks. Community detection can be achieved by fitting AGM to a given network, which was shown to outperform some state-of-the-art algorithms. Table 2 compares the performance, measured with a 3-class ($\mathcal{C} = \{\{1\}, \{2\}, \{1, 2\}\}$) classification accuracy up to global permutation $1 \leftrightarrow 2$. GNN, LGNN, LGNN-S and GAT yield similar results and outperform AGMfit. It further illustrates the benefits of data-driven models that strike the right balance between expressivity and structural design.

|  | train/test | Avg |V| | Avg |E| |  | GNN | LGNN | LGNN-S | GAT | AGMfit |
|---|---|---|---|---|---|---|---|---|---|
| Amazon | 805/142 | 60 | 161 | Avg. | 0.97 | 0.96 | 0.97 | 0.95 | 0.90 |
|  |  |  |  | Std. Dev. | 0.12 | 0.13 | 0.11 | 0.13 | 0.13 |
| DBLP | 4163/675 | 26 | 77 | Avg. | 0.90 | 0.90 | 0.89 | 0.88 | 0.79 |
|  |  |  |  | Std. Dev. | 0.13 | 0.13 | 0.13 | 0.13 | 0.18 |
| Youtube | 20000/1242 | 93 | 201 | Avg. | 0.91 | 0.92 | 0.91 | 0.90 | 0.59 |
|  |  |  |  | Std. Dev. | 0.11 | 0.11 | 0.11 | 0.13 | 0.16 |

Table 2: Comparison of the node classification accuracy by different models on the three SNAP datasets. Note that the average accuracy was computed graph-wise with each graph weighted by its size, while the standard deviation was computed graph-wise with equal weights among the graphs.

## 7 CONCLUSION

In this work, we have studied data-driven approaches to supervised community detection with graph neural networks. Our models achieve comparable performance to BP in binary SBM for various SNRs, and outperform BP in the sparse regime of 5-class SBM that falls between the computational-to-statistical gap. This is made possible by considering a family of graph operators including the power graph adjacency matrices, and importantly by introducing the line graph equipped with the non-backtracking matrix. We also provided a theoretical analysis of the optimization landscapes of simplified linear GNN for community detection and showed the gap between the loss value at local and global minima are bounded by quantities related to the concentration of certain random matricies.

One word of caution is that our empirical results are inherently non-asymptotic. Whereas models trained for given graph sizes can be used for inference on arbitrarily sized graphs (owing to the parameter sharing of GNNs), further work is needed in order to understand the generalization properties as $|V|$ increases. Nevertheless, we believe our work opens up interesting questions, namely better understanding how our results on the energy landscape depend upon specific signal-to-noise ratios, or whether the network parameters can be interpreted mathematically. This could be useful in the study of computational-to-statistical gaps, where our model can be used to inquire about the form of computationally tractable approximations. Another current limitation of our model is that it presumes a fixed number of communities to be detected. Other directions of future research include the extension to the case where the number of communities is unknown and varied, or even increasing with $|V|$, as well as applications to ranking and edge-cut problems.

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

## A  PROOF OF THEOREM 5.1

For simplicity and with an abuse of notation, in the remaining part we redefine $L$ and $\tilde{L}$ in the following way, to be the negative of their original definition in the main section: $L_n(\beta) = \mathbb{E}_{X_n, Y_n} \frac{\beta^\top Y_n \beta}{\beta^\top X_n \beta}$, $\tilde{L}_n(\beta) = \mathbb{E}_{X_n, Y_n} \frac{\beta^T Y_n \beta}{\beta^T \mathbb{E} X_n \beta}$. Thus, minimizing the loss function (5) is equivalent to maximizing the function $L_n(\beta)$ redefined here.

We write the Cholesky decomposition of $\mathbb{E}X_n$ as $\mathbb{E}X_n = R_n R_n^T$, and define $A_n = R_n^{-1} Y_n (R_n^{-1})^T$, $\bar{A}_n = \mathbb{E}A_n = R_n^{-1} \mathbb{E}Y_n (R_n^{-1})^T$, $B_n = R_n^{-1} X_n (R_n^{-1})^T$, and $\Delta B_n = B_n - I_n$. Given a symmetric matrix $K \in \mathbb{R}^{M \times M}$, we let $\lambda_1(K), \lambda_2(K), ..., \lambda_M(K)$ denote the eigenvalues of $K$ in nondecreasing order.

First, we have

$$|L_n(\beta_l) - L_n(\beta_g)| \le |L_n(\beta_l) - \tilde{L}_n(\beta_l)| + |\tilde{L}_n(\beta_l) - \tilde{L}_n(\beta_g)| + |\tilde{L}_n(\beta_g) - L_n(\beta_g)| \quad (6)$$

Let us denote by $\tilde{\beta}_g$ a global minimum of the mean-field loss $\tilde{L}_n$. Taking a step further, we can extend this bound to the following one (the difference is in the second term on the right hand side):

**Lemma A.1.**

$$|L_n(\beta_l) - L_n(\beta_g)| \le |L_n(\beta_l) - \tilde{L}_n(\beta_l)| + |\tilde{L}_n(\beta_l) - \tilde{L}_n(\tilde{\beta}_g)| + |\tilde{L}_n(\beta_g) - L_n(\beta_g)| \quad (7)$$

*Proof of Lemma A.1.* We consider two separate cases: The first case is when $\tilde{L}_n(\beta_l) \ge \tilde{L}_n(\beta_g)$. Then $\tilde{L}_n(\beta_l) - \tilde{L}_n(\tilde{\beta}_g) \ge \tilde{L}_n(\beta_l) - \tilde{L}_n(\beta_g) \ge 0$, and so $|L_n(\beta_l) - L_n(\beta_g)| \le |L_n(\beta_l) - \tilde{L}_n(\beta_l)| + |\tilde{L}_n(\beta_l) - \tilde{L}_n(\tilde{\beta}_g)| + |\tilde{L}_n(\beta_g) - L_n(\beta_g)|$.

The other case is when $\tilde{L}_n(\beta_l) < \tilde{L}_n(\beta_g)$. Note that $L_n(\beta_l) \ge L_n(\beta_g)$. Then $|L_n(\beta_l) - L_n(\beta_g)| \le |L_n(\beta_l) - \tilde{L}_n(\beta_l)| + |\tilde{L}_n(\beta_g) - L_n(\beta_g)| \le |L_n(\beta_l) - \tilde{L}_n(\beta_l)| + |\tilde{L}_n(\beta_l) - \tilde{L}_n(\tilde{\beta}_g)| + |\tilde{L}_n(\beta_g) - L_n(\beta_g)|$. $\qquad\square$

Hence, to bound the "energy gap" $|L_n(\beta_l) - L_n(\beta_g)|$, if suffices to bound the three terms on the right hand side of Lemma A.1 separately. First, we consider the second term, $|\tilde{L}_n(\beta_l) - \tilde{L}_n(\tilde{\beta}_g)|$.

Let $\gamma_l = R_n^T \beta_l, \gamma_g = R_n^T \beta_g$ and $\tilde{\gamma}_g = R_n^T \tilde{\beta}_g$. Define $S_n(\gamma) = L_n(R_n^{-T}\gamma)$ and $\tilde{S}_n(\gamma) = \tilde{L}_n(R_n^{-T}\gamma)$, for any $\gamma \in \mathbb{R}^M$. Thus, we apply a change-of-variable and try to bound $|\tilde{S}_n(\gamma_l) - \tilde{S}_n(\tilde{\gamma}_g)|$.

Since $\beta_l$ is a local maximum of $L_n$, $\lambda_1(\nabla^2 L_n(\beta_l)) \le 0$. Since $\nabla^2 S_n(\gamma_l) = R_n^{-1}\nabla^2 L_n(\beta_l)R_n^{-T}$, where $R_n$ is invertible, we know that $\lambda_1(\nabla^2 S_n(\gamma_l)) \le 0$, thanks to the following lemma:

**Lemma A.2.** *If $R, Q \in \mathbb{R}^{M \times M}$, $R$ is invertible, $Q$ is symmetric and $\lambda_q > 0$ is an eigenvalue of $Q$, then $\lambda_1(RQR^T) \ge \lambda \cdot \lambda_M(RR^T)$*

*Proof of Lemma A.2.* Say $Qw = \lambda w$ for some vector $w \in \mathbb{R}^M$. Let $v = R^{-T}w$. Then $v^T(RQR^T)v = w^T Q w = \lambda\|w\|^2$. Note that $\|w\|^2 = v^T RR^T v \ge \|v\|^2\lambda_M(RR^T)$. Hence $\lambda_1(RQR^T) \ge \frac{v^T(RQR^T)v}{\|v\|^2} \ge \frac{\lambda\|w\|^2}{\|w\|^2/\lambda_M(RR^T)} \ge \lambda \cdot \lambda_M(RR^T)$ $\qquad\square$

Since $\nabla^2 S_n(\gamma_l) = \nabla^2 \tilde{S}_n(\gamma_l) + (\nabla^2 S_n(\gamma_l) - \nabla^2 \tilde{S}_n(\gamma_l))$, there is $0 \ge \lambda_1(\nabla^2 S_n(\gamma_l)) \ge \lambda_1(\nabla^2 \tilde{S}_n(\gamma_l)) - \|\nabla^2 S_n(\gamma_l) - \nabla^2 \tilde{S}_n(\gamma_l)\|$. Hence,

$$\lambda_1(\nabla^2 \tilde{S}_n(\gamma_l)) \le \|\nabla^2 S_n(\gamma_l) - \nabla^2 \tilde{S}_n(\gamma_l)\| \quad (8)$$

Next, we relate the left hand side of the inequality above to $\cos(\gamma_l, \tilde{\gamma}_g)$, thereby obtaining an upper bound on $[1 - \cos^2(\gamma_l, \tilde{\gamma}_g)]$, which will then be used to bound $|\tilde{S}_n(\gamma_l) - \tilde{S}_n(\tilde{\gamma}_g)|$.

**Lemma A.3.** $\forall \gamma \in \mathbb{R}^d$,

$$\lambda_1(\nabla^2 \tilde{S}_n(\gamma)) \ge \frac{2}{\|\gamma\|^2}\{[1 - \cos^2(\gamma, \tilde{\gamma}_g)] \cdot [\lambda_1(\bar{A}_n) - \lambda_2(\bar{A}_n)] - 2\|\gamma\| \cdot \|\nabla\tilde{S}_n(\gamma)\|\}$$

*Proof of Lemma A.3.*

$$\begin{aligned}
\nabla^2 \tilde{S}_n(\gamma) &= 2\mathbb{E}\left[\frac{(\gamma^T\gamma)A_n - (\gamma^T A_n\gamma)I}{(\gamma^T\gamma)^2} + \frac{4(\gamma^T A_n\gamma)\gamma\gamma^T - 4(\gamma^T\gamma)A_n\gamma\gamma^T}{(\gamma^T\gamma)^3}\right] \\
&= 2\mathbb{E}\left[\frac{(\gamma^T\gamma)A_n - (\gamma^T A_n\gamma)I}{(\gamma^T\gamma)^2} + \frac{4[(\gamma^T\gamma)A_n - (\gamma^T A_n\gamma)I]\gamma\gamma^T}{(\gamma^T\gamma)^3}\right] \quad (9) \\
&= 2\left[\frac{(\gamma^T\gamma)\bar{A}_n - (\gamma^T\bar{A}_n\gamma)I}{(\gamma^T\gamma)^2} + \frac{4[(\gamma^T\gamma)\bar{A}_n - (\gamma^T\bar{A}_n\gamma)I]\gamma\gamma^T}{(\gamma^T\gamma)^3}\right]
\end{aligned}$$

Thus, if we define $Q_1 = (\gamma^T\gamma)[(\gamma^T\gamma)\bar{A}_n - (\gamma^T\bar{A}_n\gamma)I]$, $Q_2 = 4[(\gamma^T\gamma)\bar{A}_n - (\gamma^T\bar{A}_n\gamma)I]\gamma\gamma^T$, we have

$$\nabla^2 \tilde{S}_n(\gamma) = \frac{2}{\|\gamma\|^6}(Q_1 - Q_2) \quad (10)$$

To bound $\lambda_1(\nabla^2 \tilde{S}_n(\gamma))$, we bound $\lambda_1(Q_1)$ and $\|Q_2\|$ as follows:

Since $\bar{A}_n$ is symmetric, let $\hat{\gamma}_1, \dots \hat{\gamma}_M$ be the orthonormal eigenvectors of $\bar{A}_n$ corresponding to nonincreasing eigenvalues $l_1, \dots l_M$. Note that the global minimum satisfies $\tilde{\gamma}_g = \pm\hat{\gamma}_1$. Write $\gamma = \sum_{i=1}^{M} \alpha_i \hat{\gamma}_i$, and let $\bar{\alpha}_i = \frac{\alpha_i}{\sqrt{\sum_{i=1}^{M} \alpha_i^2}}$. Then $|\cos(\gamma, \tilde{\gamma}_g)| = |\cos(\gamma, \hat{\gamma}_1)| = |\bar{\alpha}_1|$.

Then,

$$\begin{aligned}
\lambda_1(Q_1) =& (\gamma^T\gamma) \left[ l_1 \sum_{i=1}^{M} \alpha_i^2 - \sum_{i=1}^{M} l_i \alpha_i^2 \right] \\
\geq& (\gamma^T\gamma) \left[ \left( (\sum_{i=1}^{M} \alpha_i^2) - \alpha_1^2 \right) (l_1 - l_2) \right] \\
=& (\gamma^T\gamma)^2 [(1 - \bar{\alpha}_1^2)(l_1 - l_2)]
\end{aligned} \tag{11}$$

To bound $\|Q_2\|$:

$$[(\gamma^T\gamma)\bar{A}_n - (\gamma^T\bar{A}_n\gamma)I]\gamma = \sum_{k=1}^{M} \left[ l_k \sum_{i=1}^{M} \alpha_i^2 - \sum_{i=1}^{M} l_i \alpha_i^2 \right] \alpha_k \hat{\gamma}_k \tag{12}$$

Note that given vectors $v, w \in \mathbb{R}^M$,

$$\|v \cdot w^T\| = |v^T w|$$

Therefore,

$$\begin{aligned}
\|Q_2\| =& 4 \left| \left( \sum_{k=1}^{M} \alpha_k \hat{\gamma}_k \right)^T \left( \sum_{k=1}^{M} [l_k(\sum_{i=1}^{M} \alpha_i^2) - (\sum_{i=1}^{M} l_i \alpha_i^2)]\alpha_k \hat{\gamma}_k \right) \right| \\
=& 4 \left| \frac{(\gamma^T\gamma)^2}{2} \gamma^T \nabla\tilde{S}(\gamma) \right| \\
\leq& 2(\gamma^T\gamma)^2 \|\gamma\| \|\nabla\tilde{S}(\gamma)\|
\end{aligned} \tag{13}$$

Thus,

$$\begin{aligned}
\lambda_1(Q_1 - Q_2) \geq& \lambda_1(Q_1) - \|Q_2\| \\
\geq& (\gamma^T\gamma)^2 ([(1 - \bar{\alpha}_1^2)(l_1 - l_2)] - 2\|\gamma\| \|\nabla_\gamma S(\gamma)\|)
\end{aligned} \tag{14}$$

This yields the desired lemma. $\qquad\square$

Combining inequality 8 and Lemma A.3, we get

$$1 - \cos^2(\gamma_l, \tilde{\gamma}_g) \leq \frac{2\|\gamma_l\| \cdot \|\nabla\tilde{S}_n(\gamma_l)\| + \frac{\|\gamma_l\|^2}{2} \|\nabla^2 S_n(\gamma_l) - \nabla^2\tilde{S}_n(\gamma_l)\|}{\lambda_1(\bar{A}_n) - \lambda_2(\bar{A}_n)} \tag{15}$$

Thus, to bound the angle between $\gamma_l$ and $\tilde{\gamma}_g$, we can aim to bound $\|\nabla\tilde{S}_n(\gamma_l)\|$ and $\|\nabla^2 S_n(\gamma_l) - \nabla^2\tilde{S}_n(\gamma_l)\|$ as functions of the quantities $\mu_n$, $\nu_n$ and $\delta_n$.

**Lemma A.4.**

$$\|\gamma_l\| \cdot \|\nabla\tilde{S}_n(\gamma_l)\| \leq 2\mu_n \nu_n \delta_n (1 + 3\nu_n + \delta\nu_n) \tag{16}$$

*Proof of Lemma A.4.*

$$\nabla S_n(\gamma) = 2\mathbb{E}\frac{A_n\gamma}{\gamma^T B_n\gamma} - 2\mathbb{E}\frac{(\gamma^T A_n\gamma)B_n\gamma}{(\gamma^T B_n\gamma)^2} \tag{17}$$

$$\nabla\tilde{S}_n(\gamma) = 2\mathbb{E}\frac{A_n\gamma}{\gamma^T\gamma} - 2\mathbb{E}\frac{(\gamma^T A_n\gamma)\gamma}{(\gamma^T\gamma)^2} \tag{18}$$

Combining equations 17 and 18, we get

$$\nabla S_n(\gamma) - \nabla \tilde{S}_n(\gamma) = \mathbb{E}\left[\frac{2(\gamma^T\gamma - \gamma^T B_n\gamma)A_n\gamma}{(\gamma^T B_n\gamma)(\gamma^T\gamma)} - \frac{2(\gamma^T A_n\gamma)[(\gamma^T\gamma)^2 B_n\gamma - (\gamma^T B_n\gamma)^2\gamma]}{(\gamma^T B_n\gamma)^2(\gamma^T\gamma)^2}\right] \tag{19}$$

Since $\nabla S_n(\gamma_l) = 0$, we have

$$\begin{aligned}
\|\nabla \tilde{S}_n(\gamma_l)\| &= \left\|\mathbb{E}\left[\frac{2(\gamma_l^T\gamma_l - \gamma_l^T B_n\gamma_l)A_n\gamma_l}{(\gamma_l^T B_n\gamma_l)(\gamma_l^T\gamma_l)} - \frac{2(\gamma_l^T A_n\gamma_l)[(\gamma_l^T\gamma_l)^2 B_n\gamma_l - (\gamma_l^T B_n\gamma_l)^2\gamma_l]}{(\gamma_l^T B_n\gamma_l)^2(\gamma_l^T\gamma_l)^2}\right]\right\| \\
&\leq \frac{2}{\|\gamma_l\|}\mathbb{E}\left[\frac{|\lambda_1(A_n)|\|\Delta B_n\|}{|\lambda_M(B_n)|} + 3\frac{|\lambda_1(A_n)|\|\Delta B_n\|}{\lambda_M^2(B_n)} + \frac{|\lambda_1(A_n)|\|\Delta B_n\|^2}{\lambda_M^2(B_n)}\right]
\end{aligned} \tag{20}$$

Then, by the generalized Hölder's inequality,

$$\begin{aligned}
\|\nabla \tilde{S}_n(\gamma_l)\| \leq &\frac{2}{\|\gamma_l\|}\Bigg[\left(\mathbb{E}|\lambda_1(A_n)|^3\mathbb{E}\|\Delta B_n\|^3\mathbb{E}\frac{1}{|\lambda_M(B_n)|^3}\right)^{\frac{1}{3}} \\
&+ 3\left(\mathbb{E}|\lambda_1(A_n)|^3\mathbb{E}\|\Delta B_n\|^3\mathbb{E}\frac{1}{|\lambda_M(B_n)|^6}\right)^{\frac{1}{3}} \\
&+ \left(\mathbb{E}|\lambda_1(A_n)|^3\mathbb{E}\|\Delta B_n\|^6\mathbb{E}\frac{1}{|\lambda_M(B_n)|^6}\right)^{\frac{1}{3}}\Bigg].
\end{aligned} \tag{21}$$

Hence, written in terms of the quantities $\mu_n$, $\nu_n$ and $\delta_n$, we have

$$\begin{aligned}
\|\gamma_l\| \cdot \|\nabla \tilde{S}_n(\gamma_l)\| &\leq 2(\mu_n\nu_n\delta_n + 3\mu_n\nu_n^2\delta_n + \mu_n\delta_n^2\nu_n^2) \\
&= 2\mu_n\nu_n\delta_n(1 + 3\nu_n + \delta\nu_n)
\end{aligned} \tag{22}$$

$\square$

**Lemma A.5.** *With $\delta_n = (\mathbb{E}\|\Delta B_n\|^6)^{\frac{1}{6}}$, $\mathbb{E}|\lambda_1(B_n)|^6 \leq 64 + 63\delta_n^6$*

*Proof of Lemma A.5.*

$$\begin{aligned}
\mathbb{E}|\lambda_1(B_n)|^6 &= \mathbb{E}\|B_n\|^6 \\
&= \mathbb{E}\|I + \Delta B_n\|^6 \\
&\leq \mathbb{E}(\|I\| + \|\Delta B_n\|)^6 \\
&= \mathbb{E}(1 + \|\Delta B_n\|)^6
\end{aligned} \tag{23}$$

Note that

$$gma \qquad \mathbb{E}(1 + X)^6 = \mathbb{E}X^6 + 6\mathbb{E}X^5 + 15\mathbb{E}X^4 + 20\mathbb{E}X^3 + 15\mathbb{E}X^2 + 6\mathbb{E}X + 1 \tag{24}$$

and for $k \in \{1, 2, 3, 4, 5\}$, if $X$ is a nonnegative random variable,

$$\begin{aligned}
\mathbb{E}X^k &= \mathbb{1}_{X>1}\mathbb{E}X^k + \mathbb{1}_{X\leq 1}\mathbb{E}X^k \\
&\leq 1 + \mathbb{1}_{X\leq 1}\mathbb{E}X^6 \\
&\leq 1 + \mathbb{E}X^6
\end{aligned} \tag{25}$$

Therefore, $\mathbb{E}|\lambda_1(B_n)|^6 \leq 64 + 63\mathbb{E}\|\Delta B_n\|^6$. $\square$

From now on, for simplicity, we introduce $\delta_n' = (64 + 63\delta_n^6)^{\frac{1}{6}}$, as a function of $\delta_n$.

**Lemma A.6.** $\forall \gamma \in \mathbb{R}^M$,

$$\begin{aligned}
\|\gamma_l\|^2 \cdot \|\nabla^2 S_n(\gamma) - \nabla^2 \tilde{S}_n(\gamma)\| \leq &\mu_n\nu_n\delta_n(10 + 14\nu_n + 2\delta_n\nu_n + 16\nu_n^2 + 16\delta_n'\nu_n \\
&+ 8\delta_n'\nu_n^2 + 8\delta_n'\nu_n + 8\delta_n\delta_n'\nu)
\end{aligned} \tag{26}$$

*Proof of Lemma A.6.*

$$\nabla^2 S_n(\gamma) - \nabla^2 \tilde{S}_n(\gamma) = 2\mathbb{E}[H_1] - 2\mathbb{E}[H_2] + 8\mathbb{E}[H_2] - 8\mathbb{E}[H_4] \tag{27}$$

where

$$H_1 = \frac{(\gamma^T \gamma) A_n - (\gamma^T B_n \gamma) A_n}{(\gamma^T B_n \gamma)(\gamma^T \gamma)} \tag{28}$$

$$H_2 = \frac{(\gamma^T A_n \gamma)[(\gamma^T \gamma)^2 B_n - (\gamma^T B_n \gamma)^2] I)}{(\gamma^T B \gamma)^2 (\gamma^T \gamma)^2} \tag{29}$$

$$H_3 = \frac{(\gamma^T A_n \gamma)[(\gamma^T \gamma)^3 B_n \gamma \gamma^T B_n^T - (\gamma^T B_n \gamma)^3 \gamma \gamma^T]}{(\gamma^T B_n \gamma)^3 (\gamma^T \gamma)^3} \tag{30}$$

$$H_4 = \frac{(\gamma^T \gamma)^2 A_n \gamma \gamma^T B_n - (\gamma^T B_n \gamma)^2 A \gamma \gamma^T}{(\gamma^T B_n \gamma)^2 (\gamma^T \gamma)^2} \tag{31}$$

Thus, $\|\nabla^2 S_n(\gamma) - \nabla^2 \tilde{S}_n(\gamma)\| \leq 2\mathbb{E}\|H_1\| + 2\mathbb{E}\|H_2\| + 8\mathbb{E}\|H_3\| + 8\mathbb{E}\|H_4\|$, and we try to bound each term on the right hand side separately.

For the first term, there is

$$\|H_1\| \leq \frac{1}{\|\gamma\|^2} \frac{\|\Delta B_n\| |\lambda_1(A_n)|}{|\lambda_M(B_n)|} \tag{32}$$

Applying generalized Hölder's inequality, we obtain

$$\|\gamma\|^2 \cdot \mathbb{E}\|H_1\| \leq \left( \mathbb{E} \frac{1}{|\lambda_M(B_n)|^3} \right)^{\frac{1}{3}} (\mathbb{E}|\lambda_1(A_n)|^3)^{\frac{1}{3}} (\mathbb{E}\|\Delta B_n\|^3)^{\frac{1}{3}} \tag{33}$$
$$\leq \mu_n \nu_n \delta_n .$$

For the second term, there is

$$H_2 = \frac{(\gamma^T A_n \gamma)[(\gamma^T \gamma)^2 \Delta B_n - 2(\gamma^T \gamma)(\gamma^T \Delta B_n \gamma) I - (\gamma^T \Delta B_n \gamma)^2 I]}{(\gamma^T B_n \gamma)^2 (\gamma^T \gamma)^2} \tag{34}$$

Hence,

$$\|H_2\| \leq \frac{1}{\|\gamma\|^2} \frac{1}{\lambda_M^2(B_n)} |\lambda_1(A_n)| (3\|\Delta B_n\| + \|\Delta B_n\|^2) \tag{35}$$

Applying generalized Hölder's inequality, we obtain

$$\|\gamma\|^2 \cdot \mathbb{E}\|H_2\| \leq \left( \mathbb{E} \frac{3}{|\lambda_M(B_n)|^6} \right)^{\frac{1}{3}} (\mathbb{E}|\lambda_1(A_n)|^3)^{\frac{1}{3}} (\mathbb{E}\|\Delta B_n\|^3)^{\frac{1}{3}}$$
$$+ \left( \mathbb{E} \frac{3}{|\lambda_M(B_n)|^6} \right)^{\frac{1}{3}} (\mathbb{E}|\lambda_1(A_n)|^3)^{\frac{1}{3}} (\mathbb{E}\|\Delta B_n\|^6)^{\frac{1}{3}} \tag{36}$$
$$\leq \mu_n \nu_n \delta_n (3\nu_n + \delta_n \nu_n)$$

For $H_3$, note that

$$(\gamma^T \gamma)^3 B_n \gamma \gamma^T B_n^T - (\gamma^T B_n \gamma)^3 \gamma \gamma^T = (\gamma^T \gamma)^3 (B_n - I) \gamma \gamma^T B_n + (\gamma^T \gamma)^3 \gamma \gamma^T (B_n - I)$$
$$+ [(\gamma^T \gamma)^3 - (\gamma^T B_n \gamma)^3] \gamma \gamma^T$$
$$= (\gamma^T \gamma)^3 \Delta B_n \gamma \gamma^T B_n + (\gamma^T \gamma)^3 \gamma \gamma^T \Delta B_n$$
$$+ [(\gamma^T B_n \gamma)^2 (-\gamma^T \Delta B_n \gamma) \gamma \gamma^T + (\gamma^T B_n \gamma)(-\gamma^T \Delta B_n \gamma) \gamma \gamma^T$$
$$+ (-\gamma^T \Delta B_n \gamma) \gamma \gamma^T] \tag{37}$$

Hence,

$$
\begin{aligned}
H_3 =(\gamma^T A_n \gamma)\Big[ &\frac{(\gamma^T\gamma)^3 \Delta B_n \gamma\gamma^T B_n + (\gamma^T\gamma)^3\gamma\gamma^T\Delta B_n + (-\gamma^T\Delta B_n\gamma)\gamma\gamma^T}{(\gamma^T B_n\gamma)^3(\gamma^T\gamma)^3} \\
&+ \frac{(-\gamma^T\Delta B_n\gamma)\gamma\gamma^T}{(\gamma^T B_n\gamma)^2(\gamma^T\gamma)} + \frac{(-\gamma^T\Delta B_n\gamma)\gamma\gamma^T}{(\gamma^T B_n\gamma)(\gamma^T\gamma)^2}\Big]
\end{aligned}
\tag{38}
$$

Thus,

$$
\|H_3\| \le \frac{|\lambda_1(A_n)|}{\|\gamma\|^2}\left[\frac{1}{|\lambda_M^3(B_n)|}(\|\Delta B_n\||\lambda_1(B_n)| + 2\|\Delta B_n\|) + \frac{1}{\lambda_M^2(B_n)}\|\Delta B_n\| + \frac{1}{|\lambda_M(B_n)|}\|\Delta B_n\|\right]
\tag{39}
$$

Applying generalized Hölder's inequality, we obtain

$$
\begin{aligned}
\|\gamma\|^2 \cdot \mathbb{E}\|H_3\| \le &\left(\mathbb{E}\frac{1}{|\lambda_M(B_n)|^6}\right)^{\frac{1}{2}}(\mathbb{E}|\lambda_1(A_n)|^6)^{\frac{1}{6}}(\mathbb{E}\|\Delta B_n\|^6)^{\frac{1}{6}}(\mathbb{E}|\lambda_1(B_n)|^6)^{\frac{1}{6}} \\
&+ 2\left(\mathbb{E}\frac{1}{|\lambda_M(B_n)|^6}\right)^{\frac{1}{2}}(\mathbb{E}|\lambda_1(A_n)|^3)^{\frac{1}{3}}(\mathbb{E}\|\Delta B_n\|^6)^{\frac{1}{6}} \\
&+ \left(\mathbb{E}\frac{1}{|\lambda_M(B_n)|^6}\right)^{\frac{1}{3}}(\mathbb{E}|\lambda_1(A_n)|^3)^{\frac{1}{3}}(\mathbb{E}\|\Delta B_n\|^3)^{\frac{1}{3}} \\
&+ \left(\mathbb{E}\frac{1}{|\lambda_M(B_n)|^3}\right)^{\frac{1}{3}}(\mathbb{E}|\lambda_1(A_n)|^3)^{\frac{1}{3}}(\mathbb{E}\|\Delta B_n\|^3)^{\frac{1}{3}} \\
\le &\mu_n\nu_n\delta_n(\delta_n'\nu_n^2 + 2\nu_n^2 + \nu_n + 1)
\end{aligned}
\tag{40}
$$

For the last term,

$$
H_4 = \frac{[-2(\gamma^T\gamma)(\gamma^T\Delta B_n\gamma)I - (\gamma^T\Delta B_n\gamma)^2 I]A_n\gamma\gamma^T B_n + (\gamma^T B_n\gamma)^2 A_n\gamma\gamma^T\Delta B_n}{(\gamma^T B_n\gamma)^2(\gamma^T\gamma)^2}
\tag{41}
$$

Thus,

$$
\|H_4\| \le \frac{1}{\|\gamma\|^2}\left[\frac{1}{\lambda_M^2(B_n)}(2\|\Delta B_n\| + \|\Delta B_n\|^2)|\lambda_1(A_n)||\lambda_1(B_n)| + \frac{1}{\lambda_M^2(B_n)}|\lambda_1^2(B_n)||\lambda_1(A_n)|\|\Delta B_n\|\right]
\tag{42}
$$

Applying generalized Hölder's inequality, we obtain

$$
\begin{aligned}
\|\gamma\|^2 \cdot \mathbb{E}\|H_4\| \le &2\left(\mathbb{E}\frac{1}{|\lambda_M(B_n)|^6}\right)^{\frac{1}{3}}(\mathbb{E}|\lambda_1(A_n)|^3)^{\frac{1}{3}}(\mathbb{E}\|\Delta B_n\|^6)^{\frac{1}{6}}(\mathbb{E}|\lambda_1(B_n)|^6)^{\frac{1}{6}} \\
&+ \left(\mathbb{E}\frac{1}{|\lambda_M(B_n)|^6}\right)^{\frac{1}{3}}(\mathbb{E}|\lambda_1(A_n)|^6)^{\frac{1}{6}}(\mathbb{E}\|\Delta B_n\|^6)^{\frac{1}{3}}(\mathbb{E}|\lambda_1(B_n)|^6)^{\frac{1}{6}} \\
&+ \left(\mathbb{E}\frac{1}{|\lambda_M(B_n)|^6}\right)^{\frac{1}{3}}(\mathbb{E}|\lambda_1(A_n)|^6)^{\frac{1}{6}}(\mathbb{E}\|\Delta B_n\|^6)^{\frac{1}{6}}(\mathbb{E}|\lambda_1(B_n)|^6)^{\frac{1}{3}} \\
\le &\mu_n\nu_n\delta_n(2\nu_n\delta_n' + \delta_n\delta_n'\nu_n + {\delta_n'}^2\nu_n)
\end{aligned}
\tag{43}
$$

Therefore, summing up the bounds above, we obtain

$$
\begin{aligned}
\|\gamma_l\|^2 \cdot \|\nabla^2 S_n(\gamma) - \nabla^2\tilde{S}_n(\gamma)\| \le &\mu_n\nu_n\delta_n(10 + 14\nu_n + 2\delta_n\nu_n + 16\nu_n^2 + 16\delta_n'\nu_n \\
&+ 8\delta_n'\nu_n^2 + 8\delta_n'\nu_n + 8\delta_n\delta_n'\nu)
\end{aligned}
\tag{44}
$$

Hence, combining inequality 15, Lemma A.4 and Lemma A.6, we get

$$
\begin{aligned}
1 - \cos^2(\gamma_l, \tilde{\gamma}_g) \le &\eta_n[4\mu_n\nu_n\delta_n(1 + 3\nu_n\delta_n\mu_n) + \frac{1}{2}\mu_n\nu_n\delta_n(10 + 14\nu_n + 2\delta_n\nu_n + 16\nu_n^2 \\
&+ 16\delta_n'\nu_n + 8\delta_n'\nu_n^2 + 8\delta_n'\nu_n + 8\delta_n\delta_n'\nu)] \\
= &\mu_n\nu_n\delta_n\eta_n(9 + 19\nu_n + 5\delta_n\nu_n + 8\nu_n^2 + 8\delta_n'\nu_n + 4\delta_n'\nu_n^2 + 4\delta_n'\nu_n + 4\delta_n\delta_n'\nu_n)
\end{aligned}
\tag{45}
$$

For simplicity, we define $C(\delta_n, \nu_n) = 9 + 19\nu_n + 5\delta_n\nu_n + 8\nu_n^2 + 8\delta_n'\nu_n + 4\delta_n'\nu_n^2 + 4\delta_n'\nu_n + 4\delta_n\delta_n'\nu_n$. Thus,

$$1 - \cos^2(\gamma_l, \tilde{\gamma}_g) \le \mu_n\nu_n\delta_n\eta_n C(\delta_n, \nu_n) \tag{46}$$

$\square$

Following the notations in the proof of Lemma A.3, we write $\gamma_l = \sum_{i=1}^{M} \alpha_i\hat{\gamma}_i$. Note that $\tilde{\gamma}_g = \pm\hat{\gamma}_1$, and $|\cos(\gamma, \hat{\gamma}_i)| = |\bar{\alpha}_i|$. Thus,

$$\begin{aligned}
\tilde{L}_n(\beta_l) =& \tilde{S}_n(\gamma_l) \\
=& \frac{\sum_{i=1}^{M} \alpha_i^2 l_i}{\sum_{i=1}^{M} \alpha_i^2} = \sum_{i=1}^{M} \bar{\alpha}_i^2 l_i
\end{aligned} \tag{47}$$

Since $Y_n$ is positive semidefinite, $\mathbb{E}Y_n$ is also positive semidefinite, and hence $\bar{A}_n = R_n^T\mathbb{E}Y_n(R_n^{-1})^T$ is positive semidefinite as well. This means that $l_i \ge 0, \forall i \in \{1, ..., M\}$. Since $\tilde{L}_n(\tilde{\beta}_g) = \tilde{S}_n(\tilde{\gamma}_g) = \tilde{S}_n(\hat{\gamma}_1) = l_1$, there is

$$|\tilde{L}_n(\tilde{\beta}_g) - \tilde{L}_n(\beta_l)| \le (1 - \bar{\alpha}_1^2)l_1 \le (1 - \cos^2(\gamma_l, \tilde{\gamma}_g))\lambda_1(\bar{A}_n) \tag{48}$$

Next, we bound the first and the third term on the right hand side of the inequality in Lemma A.1.

**Lemma A.7.** $\forall\beta$,

$$|L_n(\beta) - \tilde{L}_n(\beta)| \le (\mathbb{E}\|\Delta B_n\|^3)^{\frac{1}{3}} \cdot (\mathbb{E}|\lambda_1(A_n)|^3)^{\frac{1}{3}} \cdot \left(\mathbb{E}\Big|\frac{1}{\lambda_M(B_n)}\Big|^3\right)^{\frac{1}{3}} \tag{49}$$

*Proof of Lemma A.7.* Let $\gamma = T_n^T\beta$.

$$\begin{aligned}
|L_n(\beta) - \tilde{L}_n(\beta)| =& S_n(\gamma) - \tilde{S}_n(\gamma) \\
=& \left|\mathbb{E}\frac{(\gamma^T\Delta B_n\gamma)(\gamma^T A_n\gamma)}{(\gamma^T B_n\gamma)(\gamma^T\gamma)}\right| \\
\le& \mathbb{E}\frac{\|\Delta B_n\||\lambda_1(A_n)|}{|\lambda_M(B_n)|}
\end{aligned} \tag{50}$$

Thus, we get the desired lemma by the generalized Hölder's inequality.

$\square$

Combining inequality 46, inequality 48 and Lemma A.7, we get

$$\begin{aligned}
|L_n(\beta_l) - L_n(\beta_g)| \le& 2(\mathbb{E}\|\Delta B_n\|^3)^{\frac{1}{3}} \cdot (\mathbb{E}|\lambda_1(A_n)|^3)^{\frac{1}{3}} \cdot \left(\mathbb{E}\Big|\frac{1}{\lambda_M(B_n)}\Big|^3\right)^{\frac{1}{3}} + (1 - \cos^2(\gamma_l, \tilde{\gamma}_g))\lambda_1(\bar{A}_n) \\
\le& 2\mu_n\nu_n\delta_n + \mu_n\nu_n\delta_n\eta_n C(\delta_n, \nu_n) \cdot \lambda_1(\bar{A}_n)
\end{aligned} \tag{51}$$

Meanwhile,

$$\begin{aligned}
|L_n(\beta_g) - \tilde{L}_n(\tilde{\beta}_g)| \le& \max\{|L_n(\beta_g) - \tilde{L}_n(\beta_g)|, |L_n(\tilde{\beta}_g) - \tilde{L}_n(\tilde{\beta}_g)|\} \\
\le& (\mathbb{E}\|\Delta B_n\|^3)^{\frac{1}{3}} \cdot (\mathbb{E}|\lambda_1(A_n)|^3)^{\frac{1}{3}} \cdot (\mathbb{E}|\frac{1}{\lambda_M(B_n)}|^3)^{\frac{1}{3}} \\
\le& \mu_n\nu_n\delta_n
\end{aligned} \tag{52}$$

Hence,

$$\begin{aligned}
L_n(\beta_g) \ge& \tilde{L}_n(\tilde{\beta}_g) - \mu_n\nu_n\delta_n \\
\ge& \lambda_1(\bar{A}_n) - \mu_n\nu_n\delta_n \\
\ge& \eta_n^{-1} - \mu_n\nu_n\delta_n
\end{aligned} \tag{53}$$

, or

$$\lambda_1(\bar{A}_n) \le L_n(\beta_g) + \mu_n \nu_n \delta_n \tag{54}$$

Therefore,

$$
\begin{aligned}
|L_n(\beta_l) - L_n(\beta_g)| \le & 2\mu_n \nu_n \delta_n + (1 - \cos^2(\gamma_l, \tilde{\gamma}_g))[L_n(\beta_g) + \mu_n \nu_n \delta_n] \\
\le & \mu_n \nu_n \delta_n [2 + \eta_n \mu_n \nu_n \delta_n C(\delta_n, \nu_n)] + \eta_n \mu_n \nu_n \delta_n C(\delta_n, \nu_n) L_n(\beta_g) \\
\le & L_n(\beta_g) \left\{ \frac{\mu_n \nu_n \delta_n [2 + \eta_n \mu_n \nu_n \delta_n C(\delta_n, \nu_n)]}{\eta_n^{-1} - \mu_n \nu_n \delta_n} + \eta_n \mu_n \nu_n \delta_n C(\delta_n, \nu_n) \right\} \\
= & \frac{2\eta_n \mu_n \nu_n \delta_n [2 + C(\delta_n, \nu_n)]}{1 - \eta_n \mu_n \nu_n \delta_n} L_n(\beta_g)
\end{aligned}
\tag{55}
$$

Hence, we have proved the theorem, with $\epsilon_{\eta_n, \mu_n, \nu_n, \delta_n} = \frac{2\eta_n \mu_n \nu_n \delta_n [2 + C(\delta_n, \nu_n)]}{1 - \eta_n \mu_n \nu_n \delta_n}$. $\square$

# B  BACKGROUND

## B.1  GRAPH MIN-CUTS AND SPECTRAL CLUSTERING

We consider graphs $G = (V, E)$, modeling a system of $N = |V|$ elements presumed to exhibit some form of community structure. The adjacency matrix $A$ associated with $G$ is the $N \times N$ binary matrix such that $A_{i,j} = 1$ when $(i, j) \in E$ and 0 otherwise. We assume for simplicity that the graphs are undirected, therefore having symmetric adjacency matrices. The community structure is encoded in a discrete label vector $s : V \to \{1, \ldots, C\}$ that assigns a community label to each node, and the goal is to estimate $s$ from observing the adjacency matrix.

In the binary case, we can set $s(i) = \pm 1$ without loss of generality. Furthermore, we assume that the communities are associative, which means two nodes from the same community are more likely to be connected than two nodes from the opposite communities. The quantity

$$\sum_{i,j} (1 - s(i)s(j)) A_{i,j}$$

measures the cost associated with *cutting* the graph between the two communities encoded by $s$, and we wish to minimize it under appropriate constraints (Newman, 2006). Note that $\sum_{i,j} A_{i,j} = s^T D s$, with $D = \text{diag}(A\mathbf{1})$ (called the degree matrix), and so the cut cost can be expressed as a positive semidefinite quadratic form

$$\min_{s(i) = \pm 1} s^T (D - A) s = s^T \Delta s$$

that we wish to minimize. This shows a fundamental connection between the community structure and the spectrum of the graph Laplacian $\Delta = D - A$, which provides a powerful and stable relaxation of the discrete combinatorial optimization problem of estimating the community labels for each node. The eigenvector of $\Delta$ associated with the smallest eigenvalue is, trivially, $\mathbb{1}$, but its Fiedler vector (the eigenvector associated with the second smallest eigenvalue) reveals important community information of the graph under appropriate conditions (Newman, 2006), and is associated with the graph conductance under certain normalization schemes (Spielman, 2015).

Given linear operator $\mathcal{L}(A)$ extracted from the graph (that we assume symmetric), we are thus interested in extracting eigenvectors at the edge of its spectrum. A particularly simple algorithm is the power iteration method. Indeed, the Fiedler vector of $\mathcal{L}(A)$ can be obtained by first extracting the leading eigenvector $v$ of $\tilde{A} = \|\mathcal{L}(A)\| \mathbb{I} - \mathcal{L}(A)$, and then iteratively compute

$$y^{(n)} = \tilde{A} w^{(n-1)} \ , \ w^{(n)} = \frac{y^{(n)} - \langle y^{(n)}, v \rangle v}{\|y^{(n)} - \langle y^{(n)}, v \rangle v\|} .$$

Unrolling power iterations and recasting the resulting model as a trainable neural network is akin to the LISTA sparse coding model, which unrolled iterative proximal splitting algorithms (Gregor & LeCun, 2010).

Despite the appeal of graph Laplacian spectral approaches, it is known that these methods fail in sparsely connected graphs (Krzakala et al., 2013) . Indeed, in such scenarios, the eigenvectors of

the graph Laplacian concentrate on nodes with dominant degrees, losing their correlation with the community structure. In order to overcome this important limitation, people have resorted to ideas inspired from statistical physics, as explained next.

## B.2 PROBABILISTIC GRAPHICAL MODELS AND BELIEF-PROPAGATION

Graphs with labels on nodes and edges can be cast as a graphical model where the aim of clustering is to optimize label agreement. This can be seen as a posterior inference task. If we simply assume the graphical model is a Markov Random Field (MRF) with trivial compatibility functions for cliques greater than 2, the probability of a label configuration $\sigma$ is given by

$$\mathbb{P}(\sigma) = \frac{1}{\mathcal{Z}} \prod_{i \in V} \phi_i(\sigma_i) \prod_{ij \in E} \psi_{ij}(\sigma_i, \sigma_j). \tag{56}$$

Generally, computing marginals of multivariate discrete distributions is exponentially hard. For instance, in the case of $\mathbb{P}(\sigma_i)$ we are summing over $|X|^{n-1}$ terms (where $X$ is the state space of discrete variables). But if the graph is a tree, we can factorize the MRF more efficiently to compute the marginals in linear time via a dynamic programming method called the sum-product algorithm, also known as belief propagation (BP). An iteration of BP is given by

$$b_{i \to j}(\sigma_i) = \frac{1}{Z_{i \to j}} \phi_i(\sigma_i) \prod_{k \in \delta i \setminus j} \sum_{\sigma_k \in X} \psi_{ik}(\sigma_i, \sigma_k) b_{k \to i}(\sigma_k). \tag{57}$$

The beliefs $(b_{i \to j}(\sigma_i))$ are interpreted as the marginal distributions of $\sigma_i$. Fixed points of BP can be used to recover marginals of the MRF above. In the case of the tree, the correspondence is exact: $\mathbb{P}_i(\sigma_i) = b_i(\sigma_i)$. Certain sparse graphs, like SBM with constant degree, are locally similar to trees for such an approximation to be successful (Mossel et al., 2014). However, convergence is not guaranteed in graphs that are not trees. Furthermore, in order to apply BP, we need a generative model and the correct parameters of the model. If unknown, the parameters can be derived using expectation maximization, further adding complexity and instability to the method since it is possible to learn parameters for which BP does not converge.

## B.3 NON-BACKTRACKING OPERATOR AND BETHE HESSIAN

The BP equations have a trivial fixed-point where every node takes equal probability in each group. Linearizing the BP equation around this point is equivalent to spectral clustering using the non-backtracking matrix (NB), a matrix defined on the directed edges of the graph that indicates whether two edges are adjacent and do not coincide. Spectral clustering using NB gives significant improvements over spectral clustering with different versions of the Laplacian matrix $L$ and the adjacency matrix $A$. High degree fluctuations drown out the signal of the informative eigenvalues in the case of A and L, whereas the eigenvalues of NB are confined to a disk in the complex plane except for the eigenvalues that correspond to the eigenvectors that are correlated with the community structure, which are therefore distinguishable from the rest.

NB matrices are still not optimal in that they are matrices on the edge set and also asymmetric, therefore unable to enjoy tools of numerical linear algebra for symmetric matrices. Saade et al. (2014) showed that a spectral method can do as well as BP in the sparse SBM using the Bethe Hessian matrix defined by $BH(r) := (r^2 - 1)I - rA + D$, where $r$ is a scalar parameter. This is due to a one-to-one correspondence between the fixed points of BP and the stationary points of the Bethe free energy (corresponding Gibbs energy of the Bethe approximation) (Saade et al., 2014). The Bethe Hessian is a scaling of the Hessian of the Bethe free energy at an extrema corresponding to the trivial fixed point of BP. Negative eigenvalues of $BH(r)$ correspond to phase transitions in the Ising model where new clusters become identifiable. The success of the spectral method using the Bethe Hessian gives a theoretical motivation for having a family of matrices including $I$, $D$ and $A$ in our GNN defined in Section 4, because in this way the GNN is capable of expressing the algorithm of performing power iteration on the Bethe Hessian. While belief propagation requires a generative model, and the spectral method using the Bethe Hessian requires the selection of the parameter $r$, whose optimal value also depends on the underlying generative model, the GNN does not need a generative model and is able to learn and then make predictions in a data-driven fashion.

### B.4 Stochastic Block Model

We briefly review the main properties needed in our analysis, and refer the interested reader to Abbe (2017) for an excellent recent review. The stochastic block model (SBM) is a random graph model denoted by $SBM(n, p, q, C)$. Implicitly there is an $F : V \to \{1, \ldots, C\}$ associated with each SBM graph, which assigns community labels to each vertex. One obtains a graph from this generative model by starting with $n$ vertices and connecting any two vertices $u, v$ independently at random with probability $p$ if $F(v) = F(u)$, and with probability $q$ if $F(v) \neq F(u)$. We say the SBM is *balanced* if the communities are the same size. Let $\bar{F}_n : V \to \{1, C\}$ be our predicted community labels for $SBM(n, p, q, C)$. We say that the $F_n$'s give *exact recovery* on a sequence $\{SBM(n, p, q)\}_n$ if $\mathbb{P}(F_n = \bar{F}_n) \to_n 1$, and give *weak recovery* or *detection* if $\exists \epsilon > 0$ such that $\mathbb{P}(|F_n - \bar{F}_n| \geq 1/k + \epsilon) \to_n 1$ (i.e $\bar{F}_n$'s do better than random guessing).

It is harder to tell communities apart if $p$ is close to $q$ (if $p = q$ we just get an Erdős Renyi random graph, which has no communities). In the two community case, It was shown that exact recovery is possible on $SBM(n, p = \frac{a \log n}{n}, q = \frac{b \log n}{n})$ if and only if $\frac{a+b}{2} \geq 1 + \sqrt{ab}$ (Mossel et al., 2014; Abbe et al., 2014). For exact recovery to be possible, $p, q$ must grow at least $O(\log n)$ or else the sequence of graphs will not be connected, and thus the vertex labels will be underdetermined. There is no information-computation gap in this regime, and so there exist polynomial time algorithms when recovery is possible (Abbe, 2017; Mossel et al., 2014)). In the sparser regime of constant degree, $SBM(n, p = \frac{a}{n}, q = \frac{b}{n})$, detection is the best we could hope for. The constant degree regime is also of most interest to us for real world applications, as most large datasets have bounded degree and are extremely sparse. It is also a very challenging regime; spectral approaches using the Laplacian in its various (un)normalized forms or the adjacency matrix, as well as semidefinite programming (SDP) methods do not work well in this regime due to large fluctuations in the degree distribution that prevent eigenvectors from concentrating on the clusters (Abbe, 2017). Decelle et al. (2011) first proposed the BP algorithm on the SBM, which was proven to yield Bayesian optimal values in Coja-Oghlan et al. (2016).

In the constant degree regime with $k$ balanced communities, the signal-to-noise ratio is defined as $SNR = (a - b)^2/(k(a + (k + 1)b))$, and the Kesten-Stigum (KS) threshold is given by $SNR = 1$ (Abbe, 2017). When $SNR > 1$, detection can be solved in polynomial time by BP (Abbe, 2017; Decelle et al., 2011). For $k = 2$, it has been shown that when $SNR < 1$, detection is not solvable, and therefore $SNR = 1$ is both the computational and the information theoretic threshold (Abbe, 2017). For $k > 4$, it has been shown that for some $SNR < 1$, there exists non-polynomial time algorithms that are able to solve the detection problem (Abbe, 2017). Furthermore, it is conjectured that no polynomial time algorithm can solve detection when $SNR < 1$, in which case a gap would exist between the information theoretic threshold and the KS threshold (Abbe, 2017).

## C Further Experiments

### C.1 Geometric Block Model

Table 3: Overlap performance (in percentage) of GNN and LGNN on graphs generated by the Geometric Block Model compared with two spectral methods

| Model | $S = 1$ | $S = 2$ | $S = 4$ |
|---|---|---|---|
| Norm. Laplacian | $1 \pm 0.5$ | $1 \pm 0.6$ | $1 \pm 1$ |
| Bethe Hessian | $18 \pm 1$ | $38 \pm 1$ | $38 \pm 2$ |
| GNN | $20 \pm 0.4$ | $39 \pm 0.5$ | $39 \pm 0.5$ |
| LGNN | $\mathbf{22 \pm 0.4}$ | $\mathbf{50 \pm 0.5}$ | $\mathbf{76 \pm 0.5}$ |

The success of belief propagation on the SBM relies on its locally hyperbolic properties, which make it tree-like with high probability. This behavior is completely different if one considers random graphs with locally Euclidean geometry. The Geometric Block Model (Sankararaman & Baccelli, 2018) is a random graph generated as follows. We start by sampling $n$ points $x_1, \ldots, x_n$ i.i.d. from a Gaussian mixture model given by means $\mu_1, \ldots \mu_k \in \mathbb{R}^d$ at distances $S$ apart and identity covariances. The label of each sampled point corresponds to which Gaussian it belongs to. We then draw an edge between two nodes $i, j$ if $\|x_i - x_j\| \leq T/\sqrt{n}$. Due to the triangle inequality, the model contains a large number of short cycles, which affects the performance of loopy belief propagation. This

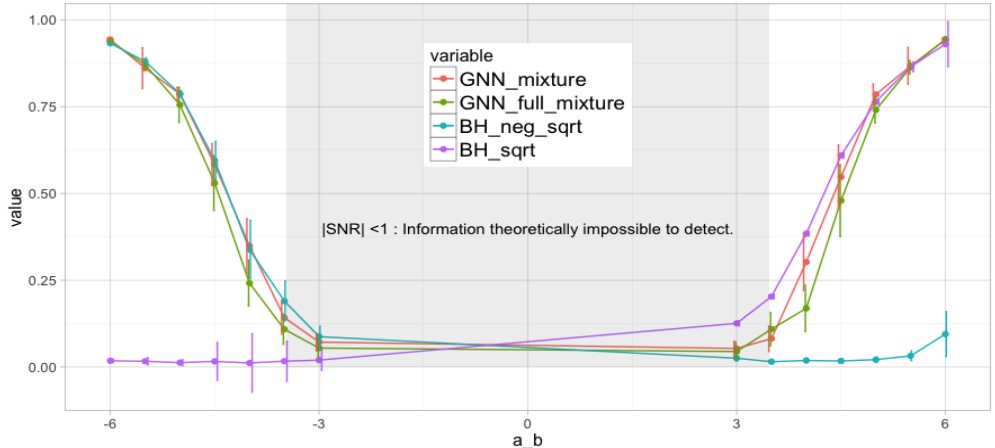

Figure 4. GNN mixture (Graph Neural Network trained on a mixture of SBM with average degree 3), GNN full mixture (GNN trained over different SNR regimes), $BH(\sqrt{\bar{d}})$ and $BH(-\sqrt{\bar{d}})$. *left:* $k = 2$. We verify that $BH(r)$ models cannot perform detection at both ends of the spectrum simultaneously.

motivates other estimation algorithms based on motif-counting that require knowledge of the model likelihood function (Sankararaman & Baccelli, 2018).

Table 3 shows the performance of GNN and LGNN on the binary GBM model, obtained with $d = 2$, $n = 500$, $T = 5\sqrt{2}$ and varying $S$, as well as the performances of two spectral methods, using respectively the normalized Laplacian and the Bethe Hessian, which approximates BP around its stationary solution. We note that LGNN model, thanks to its added flexibility and the multiscale nature of its generators, is able to significantly outperform both spectral methods as well as the baseline GNN.

## C.2 Mixture of binary SBM

We report here our experiments on the SBM mixture, generated with

$$G \sim SBM(n = 1000, p = k\bar{d} - q, q \sim \text{Unif}(0, \bar{d} - \sqrt{\bar{d}}), C = 2) ,$$

where the average degree $\bar{d}$ is either fixed constant or also randomized with $\bar{d} \sim \text{Unif}(1, t)$. Figure 4 shows the overlap obtained by our model compared with several baselines. Our GNN model is either competitive with BH or outperforms BH, which achieves the state of the art along with BP Saade et al. (2014), despite not having any access to the underlying generative model (especially in cases where GNN was trained on a mixture of SBM and thus must be able to generalize the $r$ parameter in BH). They all outperform by a wide margin spectral clustering methods using the symmetric Laplacian and power method applied to $\|BH\|I - BH$ using the same number of layers as our model. Thus GNN's ability to predict labels goes beyond approximating spectral decomposition via learning the optimal $r$ for $BH(r)$. The model architecture could allow it to learn a higher dimensional function of the optimal perturbation of the multiscale adjacency basis, as well as nonlinear power iterations, that amplify the informative signals in the spectrum.

## C.3 Ablation studies of GNN and LGNN

Compared to $f$, each of $h$, $i$ and $k$ has one fewer operator in $\mathcal{F}$, and $j$ has two fewer. We see that with the absence of $A^{(2)}$, $k$ has much worse performance than the other four, indicating the importance of the power graph adjacency matrices. Interestingly, with the absence of $I$, $i$ actually has better average accuracy than $f$. One possibly explanation is that in SBM, each node has the same expected degree, and hence $I$ may be not very far from $D$, which might make having both $I$ and $D$ in the family redundant to some extent.

Comparing GNN models $a$, $b$ and $c$, we see it is not the case that having larger $J$ will always lead to better performance. Compared to $f$, GNN models $c$, $d$ and $e$ have similar numbers of parameters but

|     |        | #layers | #features | J  | $\mathcal{F}_A$ | #parameters | Avg. | Std. Dev. |
|-----|--------|---------|-----------|----|----------------|-------------|--------|-----------|
| (a) | GNN    | 30      | 8         | 2  | $I, D, A, A^2$ | 8621        | 0.1792 | 0.0385    |
| (b) | GNN    | 30      | 8         | 4  | $I, D, A, ..., A^{(4)}$ | 12557 | 0.1855 | 0.0438    |
| (c) | GNN    | 30      | 8         | 11 | $I, D, A, ..., A^{(11)}$ | 26333 | 0.1794 | 0.0359    |
| (d) | GNN    | 30      | 15        | 2  | $I, D, A, A^{(2)}$ | 28760   | 0.1894 | 0.0388    |
| (e) | GNN    | 30      | 12        | 4  | $I, D, A, ..., A^{(4)}$ | 27273 | 0.1765 | 0.0371    |
| (f) | LGNN   | 30      | 8         | 2  | $I, D, A, A^{(2)}$ | 25482   | 0.2073 | 0.0481    |
| (g) | LGNN-L | 30      | 8         | 2  | $I, D, A, A^{(2)}$ | 25482   | 0.1822 | 0.0395    |
| (h) | LGNN   | 30      | 8         | 2  | $I, A, A^{(2)}$ | 21502      | 0.1981 | 0.0529    |
| (i) | LGNN   | 30      | 8         | 2  | $D, A, A^{(2)}$ | 21502      | 0.2212 | 0.0581    |
| (j) | LGNN   | 30      | 8         | 2  | $A, A^{(2)}$    | 17622      | 0.1954 | 0.0441    |
| (k) | LGNN   | 30      | 8         | 1  | $I, D, A$       | 21502      | 0.1673 | 0.0437    |
| (l) | LGNN-S | 30      | 8         | 2  | $I, D, A, A^{(2)}$ | 21530   | 0.1776 | 0.0398    |

Table 4: The effects of different architectures and choices of the operator family for GNN and LGNN, as demonstrated by their performance on the 5-class dissociative SBM experiments with the exact setup as in Section 6.2. For LGNN, $\mathcal{F}_B$ is the same as $\mathcal{F}_A$ except for changing $A$ to $B$.

all achieve worse average test accuracy, indicating that the line graph structure is essential for the good performance of LGNN in this experiment. In addition, $l$ also performs worse than $f$, indicating the significance of the non-backtracking line graph compared to the symmetric line graph.

