# OpenReview forum: "Supervised Community Detection with Line Graph Neural Networks"
_ICLR.cc/2019/Conference_

### Official Review · AnonReviewer1 · 2018-10-28
**An impressive piece of work opening the exciting possibility of discovering optimal algorithms with machine learning. A couple of misleading statements to be adjusted.**

**Rating:** 8
**Confidence:** 4

**Review:**

This paper presents a study of the community detection problem via graph neural networks. The presented results open the possibility that neural networks are able to discover the optimal algorithm for a given task. This is rather convincingly demonstrated on the example of the stochastic block model, where the optimal performance is known (for 2 symmetric groups) or strongly conjectured (for more groups). The method is rather computationally demanding, and also somewhat unrealistic in the aspect that the training examples might not be available, but for a pioneering study of this kind this is well acceptable.

Despite my overall very positive opinion, I found a couple of claims that are misleading and overall hurt the quality of the paper, and I would strongly suggest to the authors to adjust these claims:

** The method is claimed to "even improve upon current computational thresholds in hard regimes." This is misleading, because (as correctly stated in the body of the paper) the computational threshold to which the paper refers apply in the limit of large graph sizes whereas the observed improvements are for finite sizes. It is shown here that for finite sizes the present method is better than belief propagation. But this clearly does not imply that it improves the conjectured computational thresholds that are asymptotic. At best this is an interesting hypothesis for future work, not more.

** The energy landscape is analyzed "under certain simplifications and assumptions". Conclusions state "an interesting transition from rugged to simple as the size of the graphs increase under appropriate concentration conditions." This is very vague. It would be great if the paper could offer intuitive explanation of there simplifications and assumptions that is between these unclear remarks and the full statement of the theorem and the proof that I did not find simple to understand. For instance state the intuition on in which region of parameters are those results true and in which they are not.

** "multilinear fully connected neural networks whose landscape is well understood (Kawaguchi, 2016)." this is in my opinion grossly overstated. While surely that paper presents interesting results, they are set in a regime that lets a lot to be still understood about landscape of fully connected neural networks. It is restricted to specific activation functions, and the results for non-linear networks rely on unjustified simplifications, the sample complexity trade-off is not considered, etc.


Misprint: Page 2: cetain -> certain.

---

> ### Author Response · Authors · 2018-11-12
> **Response to the Review**
>
> We very much appreciate the compliments as well as the comments on the several claims in the paper.
>
> By “improving upon current computational thresholds in hard regimes,” indeed we meant to say that the results on finite-size graphs of our algorithms are better than those of belief propagation, which is known to reach the computational threshold of such problems. We will change the phrasing of the claim in the paper.
>
> “On the simplifications of the energy landscape analysis”:
> The simplifications that we made in the theoretical analysis are actually discussed in detail in section 5, including using squared cosine distance in place of cross-entropy loss, using a single feature map, removing nonlinearities, replacing spatial batch normalization by projection onto the unit l_2 ball, as well reparametrizing the network’s parameters according to the Krylov subspace generated by the set of operators. Assumptions are the four quantities defined in Theorem 5.1 are finite. It is indeed a highly interesting question under which of graphs (for example, for what regimes of the stochastic block model) these assumptions are satisfied. We don’t have theoretical results for this question yet, although it will certainly be of great interest to future work.
>
> On "multilinear fully connected neural networks whose landscape is well understood (Kawaguchi, 2016)." this is in my opinion grossly overstated.”
>
> The reviewer is correct in that the optimization landscape of deep, nonlinear neural networks is still far from understood. We were referring to the case with no activation functions (multilinear), in which the situation is much simpler. We will modify the text to make sure there is no ambiguity.

---

> > ### Comment · AnonReviewer1 · 2018-11-23
> > **Ok for the answer**
> >
> > Possibly it would help the reader, in order to connect the different parts of the paper, if the authors say in Section 5  explicitly that specifying the region of parameters for which these assumptions are satisfied for the SBM (and other models) is an open question.
> >
> > Otherwise I find the suggested adjustments satisfactory, and maintain my original rating.

---

### Official Review · AnonReviewer3 · 2018-11-05
**Interesting new take on GNN with the non-backtracking operator**

**Rating:** 9
**Confidence:** 4

**Review:**

Graph Neural Networks(GNN) are gaining traction and generating a lot of interest. In this work, the authors apply them to the community detection problem, and in particular to graphs generated from the stochastic block model. The main new contribution here is called "line graph neural network" that operate directly over the edges of the graph, using efficiently the power of the "non backtracking operator" as a spectral method for such problems.

Training such GNN on data generated from the stochastic block model and other graph generating models, the authors shows that the resulting method can be competitive on both artificial and real datasets.

This is definitely an interesting idea, and a nice contribution to GNN, that should be of interest to ICML folks.

References and citations are fine for the most part, except for one very odd exception concerning one of the main object of the paper: the non-backtracking operator itself! While discussed in many places, no references whatsoever are given for its origin in detection problems. I believe this is due to (Krzakala et al, 2013) ---a paper cited for other reasons--- and given the importance of the non-backtracking operator for this paper, this should be acknowledged explicitly.

Pro: Interesting new idea for GNN, that lead to more powerful method and open exciting direction of research. A nice theoretical analysis of the landscape of the graph.

Con:The evidence provided in Table 1 is rather weak. The hard phase is defined in terms of computational complexity (polynomial vs exponential) and therefore require tests on many different sizes.

---

> ### Author Response · Authors · 2018-11-12
> **Response**
>
> We sincerely thank the reviewer for his time and constructive comments.
>
> Regarding the reference of Krzakala et al., 2013, “Spectral redemption in clustering sparse networks”, you are correct that we should mention the fact that it introduced the non-backtracking operator for community detection. Thanks for this important remark, this is in fact a landmark paper central to our construction.
>
> “On the Computational-Statistical Gap Experiment”
> It is correct that the computational and statistical thresholds for detection are defined asymptotically, and therefore our experimental results with finite-size graphs do not contradict those thresholds. We only hoped to demonstrate the good performance of the GNN and LGNN models in these scenarios. We hypothesize two possible scenarios: either that the network is picking up finite-size effects that standard BP is unable to exploit, either that the network actually improves asymptotic detection. We are currently exploring this question and hoping to provide some answers to it. In any case, we appreciate your comment, and will modify the statement of the implication of our experimental results in the paper.

---

### Official Review · AnonReviewer2 · 2018-11-06
**an interesting and novel GNN, but somehow unclear in experiments.**

**Rating:** 6
**Confidence:** 4

**Review:**

This paper introduces a novel graph conv neural network, dubbed LGNN, that extends the conventional GNN using the line graph of edge adjacencies and a non-backtracking operator. It has a form of learning directed edge features for message-passing. An energy landscape analysis of the LGNN is also provided under linear assumptions. The performance of LGNN is evaluated on the problem of community detection, comparing with some baseline methods.

I appreciate the LGNN formulation as a reasonable and nice extension of GNN. The formulation is clearly written and properly discussed with message passing algorithms and other GNNs. Its potential hierarchical construction is also interesting, and maybe useful for large-scale graphs. In the course of reading this paper, however, I don’t find any clear reason why this paper restricts itself to community detection, rather than general node-classification problems for broader audience. It would have been more interesting if it covers other classification datasets in their experiments.

Most of the weak points of this paper lie in the experimental section.
1. The experimental sections do not have proper ablation studies, e.g., as follows.
As commented in Sec 6.3, GAT may underperform due to the absence of the degree matrix and this needs to be confirmed by running GAT with the degree term. And, as commented in footnote 4, the authors used spatial batch normalization to improve the performance of LGNN. But, it’s not clear how much it obtains for each experiment and, more importantly, whether they use the same spatial batch norm in other baselines. To make sure the actual gain of LGNN, this needs be done with some ablation studies.
2. The performance gain is not so significant compared to other simpler baselines, so the net contribution of  the line-graph extension is unclear considering the above.
3. The experimental section considers only a few number of classes (2-5) so that it’s does not show how it scales with a large number of classes. In this sense, other benchmark datasets with more classes (e.g., PPI datasets used in GAT paper) would be better.

I hope to get answers to these.

---

> ### Author Response · Authors · 2018-11-12
> **Response**
>
> Thank you very much for the constructive and high-quality comments.
>
> “…why this paper restricts itself to community detection, rather than general node-classification problems for broader audience”
>
> The reason why we restrict ourselves to community detection problems is that it is a relatively well-studied setup, for which several algorithms have been proposed, and where computational and statistical thresholds are known in several cases. In addition, synthetic datasets can be easily generated for community detection. Therefore, we think it is a good testbed for comparing different algorithms. However, it is a very good point that GNN and LGNN can be applied to other node-wise classification problems as well. We will modify the text to highlight this point.
>
> “To make sure the actual gain of LGNN, this needs be done with some ablation studies.”
>
> This is a valid suggestion. You correctly pointed out that GAT does not utilize the degree matrix directly, and so we are planning to perform ablation experiments by removing the degree matrix from GNN and LGNN.  We did add spatial batch normalization steps to the GAT and MPNN models we used, and in the experiments we found that spatial batch normalization is crucial for the performance of the models including GNN, LGNN, GAT and MPNN. The reason for this is outlined at the end of Section 4.1, in which we assimilate the spatial normalization with removing the DC component of node features, which is aligned with the eigenvector of the adjacency matrix of leading eigenvalue.
>
>
>  “The performance gain is not so significant compared to other simpler baselines, so the net contribution of the line-graph extension is unclear considering the above.”
>
> Although not all differences in the results are statistically significant (where we consider 2 sigma to be significant), we still think it is worth noting that in all of the experiments (binary SBM, 5-class dissociative SBM, GBM and SNAP data), LGNN achieved better averaged performance than all other algorithms, including GNN without line graph included. We also note that the complexity in operations/memory of using LGNN is the same as the alternative edge-learning methods we compared against, so these gains come essentially for free.
>
> "The experimental section considers only a few number of classes (2-5) so that it’s does not show how it scales with a large number of classes"
>
> This is indeed an interesting direction for future research. We will highlight this current limitation and discuss possible routes.

---

### Author Response · Authors · 2018-11-27
**updated version**

We would like to thank again our three reviewers for their time and high-quality feedback. We have integrated their comments into an updated manuscript. The main changes include:

-- ablation experiments of our GNN/LGNN architectures, in Sections 6.1 and 6.2
-- fixed several typos.
-- clarified assumptions of our landscape analysis (and mention that an open question is to study their validity in SBM models). (Section 5).
-- clarified finite-sample effects in our computational-to-statistical gap results (Section 6.2).

---

### Meta-Review · Area_Chair1 · 2018-12-14
**Good paper, accept**

**Confidence:** 4
**Recommendation:** Accept (Poster)

**Metareview:**

This paper introduces a new graph convolutional neural network, called LGNN, and applied it to solve the community detection problem. The reviewers think LGNN yields a nice and useful extension of graph CNN, especially in using the line graph of edge adjacencies and a non-backtracking operator.  The empirical evaluation shows that the new method provides a useful tool for real datasets. The reviewers raised some issues in writing and reference, for which the authors have provided clarification and modified the papers accordingly.